# SPURIOUS CORRELATION-AWARE EMBEDDING REGULARIZATION FOR WORST-GROUP ROBUSTNESS

**Subeen Park[1], Joowang Kim[2], Hakyung Lee[3], Sunjae Yoo[1], Kyungwoo Song[1]\***
[1]Yonsei University, Republic of Korea
[2]LG CNS, Republic of Korea
[3]Korea Development Bank, Republic of Korea

## ABSTRACT

Deep learning models achieve strong performance across various domains but often rely on spurious correlations, making them vulnerable to distribution shifts. This issue is particularly severe in subpopulation shift scenarios, where models struggle in underrepresented groups. While existing methods have made progress in mitigating this issue, their performance gains are still constrained. They lack a theoretical motivation connecting the embedding space representations with worst-group error. To address this limitation, we propose Spurious Correlation-Aware Embedding Regularization for Worst-Group Robustness (SCER), a novel approach that directly regularizes feature representations to suppress spurious cues. We theoretically show that worst-group error is influenced by how strongly the classifier relies on spurious versus core directions, as identified from differences in group-wise mean embeddings across domains and classes. By imposing theoretical constraints at the embedding level, SCER encourages models to focus on core features while reducing sensitivity to spurious patterns. Through systematic evaluation on multiple vision and language tasks, we show that SCER outperforms prior state-of-the-art methods in worst-group accuracy. Our code is available at https://github.com/MLAI-Yonsei/SCER.

## 1 INTRODUCTION

Deep neural architectures have transformed performance standards in many domains. However, they are still highly vulnerable to spurious correlations in real-world datasets (Yang et al., 2023a; Izmailov et al., 2022; Sagawa et al., 2020). Spurious correlations are patterns in the training data that are statistically associated with target labels but lack a genuine causal relationship. Such correlations are often the result of biases in the dataset, causing models to fail when faced with distribution shifts (Ye et al., 2024). Recent studies have shown that this issue is particularly severe in subpopulation shift scenarios, where the distribution of specific subpopulations in the test data differs from that in the training data (Sagawa et al.; Yang et al., 2023b). Models trained with Empirical Risk Minimization (ERM) tend to over-rely on these spurious correlations, as ERM minimizes the average loss without considering subpopulation imbalances (Vapnik, 2013; Izmailov et al., 2022). As a result, models rely on easily learned but non-robust features, leading to poor worst-group accuracy and unreliable performance across underrepresented subpopulations.

Numerous approaches have been developed to handle this challenge, which can be broadly grouped into several categories (Yang et al., 2023b). Subpopulation robustness methods (Sagawa et al.; Duchi & Namkoong, 2021; Liu et al., 2021; Yao et al., 2022; Deng et al., 2023) focus on improving worst-group performance through various techniques such as reweighting, multi-stage training, or specialized loss functions. Domain invariant methods (Arjovsky et al., 2019; Sun & Saenko, 2016) attempt to learn features that generalize across different domains by aligning feature distributions. Data augmentation approaches (Zhang et al., 2018) create synthetic training examples to improve generalization. Class imbalance methods (Japkowicz, 2000; Cui et al., 2019; Ren et al., 2020) address the uneven distribution of samples across classes. However, these approaches primarily influence model learning indirectly through sample reweighting or feature alignment, without explicitly

---
\*Corresponding author.

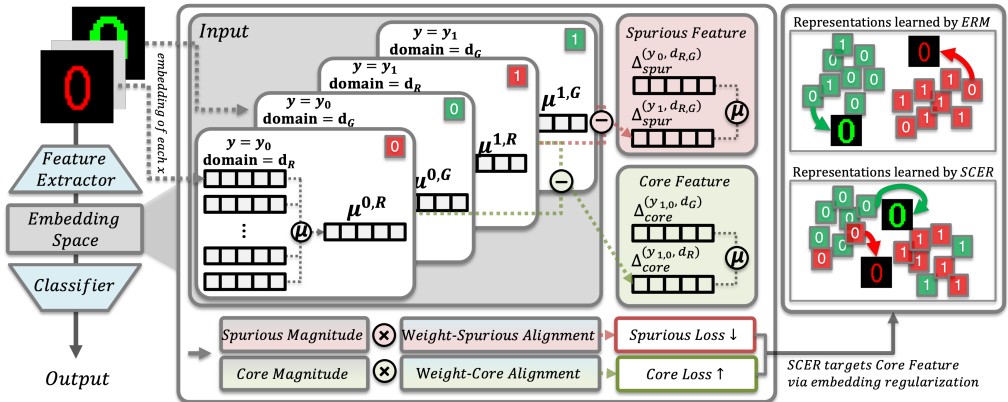

Figure 1: Overview of the Spurious Correlation-Aware Embedding Regularization (SCER) framework. Input data is encoded into embeddings, from which group-wise mean embeddings are computed for each label-domain pair. SCER decomposes these into **Spurious Features**, capturing domain differences within each class, and **Core Features**, capturing class differences within each domain. **Spurious Magnitude** and **Core Magnitude** quantify the strength of domain-driven and label-driven variations. **Weight-Spurious Alignment** and **Weight-Core Alignment** measure how strongly the classifier aligns with each component. SCER constructs Spurious Loss and Core Loss from these terms to regulate the embedding space, reducing Spurious Loss to suppress domain-specific patterns while increasing Core Loss to reinforce label-relevant signals, thereby improving robustness to domain bias.

constraining how spurious features are encoded within the embedding space. Consequently, spurious correlations may persist, limiting robustness under distribution shifts.

To address this limitation, we propose Spurious Correlation-Aware Embedding Regularization for Worst-Group Robustness (SCER), a novel method that directly regularizes feature representations to prevent reliance on spurious features. As illustrated in Figure 1, SCER operates at the embedding level, imposing explicit constraints that guide the model toward learning more robust features. In this study, we provide a new theoretical relationship between worst-group error and the structural properties of the representation space under subpopulation shift scenarios, where spurious correlations exist between domain and label. We identify two key factors that influence worst-group error through our theoretical analysis. First, for the same-label samples across different domains, we regularize their embeddings to align on essential classification features. Second, for instances sharing the same domain but with different labels, we introduce additional constraints to reduce the impact of domain-specific spurious cues. Through this approach, SCER prevents the model from overfitting to domain-level artifacts, enabling it to learn robust core features more effectively.

By structuring the model's representation space, SCER improves worst-group accuracy while maintaining strong overall performance. Through extensive experiments across image and text datasets, we demonstrate that SCER outperforms prior methods in worst-group accuracy. We also provide quantitative and qualitative evidence that embedding-level constraints yield robust feature representations even under spurious domain shifts, confirming the effectiveness of the SCER.

## 2 RELATED WORKS

### 2.1 SUBPOPULATION SHIFT

Machine learning models degrade under subpopulation shift, where the test subpopulation distributions differ from those in the training data (Yang et al., 2023b). ERM's focus on average training loss makes models susceptible to spurious correlations (Vapnik, 2013; Izmailov et al., 2022). To address this issue, various methods have been suggested, which can be broadly categorized as sample reweighting and data augmentation strategies. GroupDRO (Sagawa et al.) reduces worst-group error by dynamically upweighting samples from groups with higher loss. LfF (Bahng et al., 2020) employs a two-stage approach that first detects spurious correlations by training a biased model and then debiases a second model by reweighting the loss gradient. GIC (Han & Zou, 2024) improves the accuracy of group inference by leveraging spurious attribute associations. For data augmenta-

tion, LISA (Yao et al., 2022) enhances robustness by interpolating between subpopulation samples to encourage invariant representations. Similarly, PDE (Deng et al., 2023) gradually expands the training set, allowing models to first learn core features before incorporating more diverse samples. While prior research focuses primarily on sample reweighting or data augmentation, we propose SCER to directly regularize the embedding space and reduce reliance on spurious features. By explicitly penalizing reliance on spurious cues at the representation level, SCER significantly improves robustness across various distribution shift benchmarks.

## 2.2 EMBEDDING SPACE

Recent studies in the text and image domains have actively explored methods to refine latent spaces, aiming to improve representation quality and better capture the underlying structure among samples (Patil et al., 2023; Ming et al.). However, in the domain generalization setting, research has primarily focused on learning invariant features rather than explicitly modifying the embedding space. Previous work has attempted to mitigate unintended correlations arising from background, lighting, or dataset-specific biases (Yang et al., 2023b). Hermann & Lampinen (2020) analyzed how networks differentiate robust and spurious features, while LfF (Bahng et al., 2020) introduced a two-stage training approach to guide model learning. More recently, ElRep (Wen et al., 2025) applies norm penalties to final-layer representations with demonstrated practical effectiveness. However, these methods typically guide model learning indirectly rather than explicitly constraining spurious feature encoding in the embedding space (Patil et al., 2023; Ming et al.). In contrast to previous approaches, we propose to directly impose an explicit loss function on the embedding space, ensuring that learned representations minimize reliance on spurious correlations and improve generalization under distribution shifts. Our work addresses these theoretical gaps by providing motivation for embedding regularization and establishing clear connections to robust generalization.

## 3 METHODOLOGY

### 3.1 PROBLEM SETTING

We consider a classification problem where each data point $x \in \mathcal{X}$ is associated with a label $y \in \mathcal{Y}$ and a domain $d \in \mathcal{D}$, where $\mathcal{Y} = \{y_1, y_2, \ldots, y_m\}$ is the set of $m$ classes and $\mathcal{D} = \{d_1, d_2, \ldots, d_k\}$ is the set of $k$ domains. Each pair $(y, d) \in \mathcal{Y} \times \mathcal{D}$ defines a *subpopulation*, which may exhibit a *spurious correlation* if $d$ is not causally related to $y$ but appears disproportionately often with certain $y$ values. We aim to learn a classifier $f : \mathcal{X} \to \mathcal{Y}$, which consists of a feature extractor $f_w : \mathcal{X} \to \mathbb{R}^p$ and a classifier $f_\beta : \mathbb{R}^p \to \mathcal{Y}$, such that $f(x) = f_\beta(f_w(x))$. A standard ERM objective aims to minimize

$$\min_f \ \mathbb{E}_{(x,y) \sim \mathbb{P}}\big[\ell\big(f(x), \, y\big)\big],$$

where $\ell$ is the loss and $\mathbb{P}$ is the full data distribution. Although ERM effectively reduces *average* error, it can fail on *minority groups*, leading to large misclassification rates in those groups.

**Subpopulation Shift.** We consider a learning setting where performance should remain robust across all subpopulations defined by label-domain pairs $(y, d)$. To address imbalances among these groups, we adopt the worst-case optimization framework of Group Distributionally Robust Optimization (GroupDRO), which can be formulated as

$$\min_f \max_{q \in \mathcal{Q}} \sum_{(y,d) \in \mathcal{Y} \times \mathcal{D}} q_{(y,d)} \cdot \mathbb{E}_{(x,y,d) \sim \mathbb{P}_{y,d}}\big[\ell\big(f(x), \, y\big)\big]$$

Here, $\mathbb{P}_{y,d}$ represents the data distribution conditioned on subpopulation $(y, d)$, $q$ represents how much weight to give each subpopulation during training, and $\mathcal{Q}$ denotes the set of all possible ways to assign these weights. This minimax formulation ensures robustness across all subpopulations.

To further enhance robustness, we introduce a regularization approach motivated by the decomposition of worst-group error into spurious and core components within the embedding space. Rather than addressing group errors solely through loss optimization, we incorporate structural regularization that encourages the classifier to align with label-consistent **Core Feature** while penalizing dependence on domain-specific **Spurious Feature**. These regularization terms, formulated through directional correlations in the embedding space, are detailed in the following section.

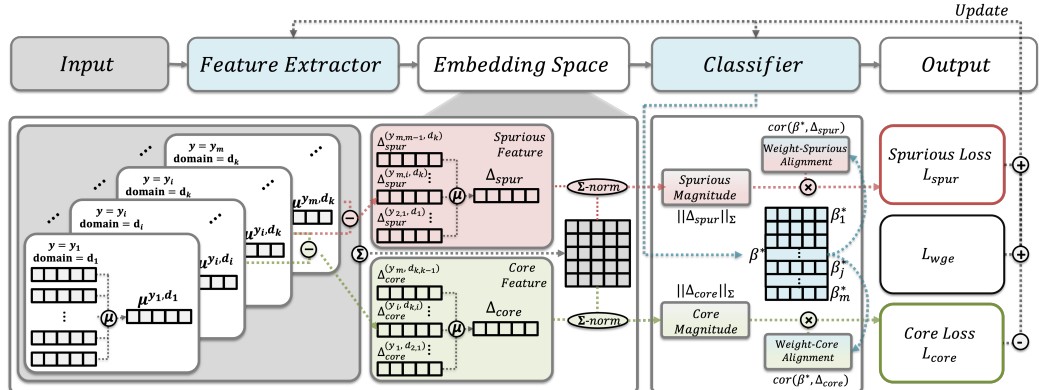

Figure 2: Overview of the SCER training framework. Input data is encoded into embeddings, from which group-wise mean embeddings are computed to derive spurious and core directions. The framework combines worst-group classification loss with embedding regularization that penalizes spurious alignment and promotes core alignment.

## 3.2 SCER

We propose SCER, which enhances the generalization performance of classifiers by decomposing the worst-group error and applying embedding-level regularization to each component, as shown in Figure 2. Let $x_{\mathrm{emb}} = f_w(x) \in \mathbb{R}^p$ denote the embedding vectors of input $x$, obtained by a feature extractor $f_w$ with parameters $w$, where $p$ is the embedding dimension. For each subpopulation defined by a label–domain pair $(y, d)$, we define its *mean embedding* as

$$\mu^{(y,d)} = \mathbb{E}_{x \sim \mathbb{P}_{y,d}}[f_w(x)] \in \mathbb{R}^p.$$

We compute mean embeddings as representative vectors for each subpopulation, enabling decomposition of worst-group error into two primary components.

**Spurious Feature.** The embedding mean difference in feature representations across different domains within the same class. Since we assume that label–domain pairs $(y, d)$ may exhibit spurious correlations, such differences should be minimized to prevent the model from overfitting to domain-specific artifacts. Formally, the spurious difference for a given class $y$ is defined as

$$\Delta_{\mathrm{spur}}^{(y,d_{i,j})} = \left| \mu^{(y,d_i)} - \mu^{(y,d_j)} \right|, \quad \forall y \in \mathcal{Y}, \ d_i, d_j \in \mathcal{D}, \ d_i \neq d_j.$$

**Core Feature.** The embedding mean difference in feature representations between different classes within the same domain. This quantity captures the actual signal that the classifier should leverage for robust prediction. Formally, the core difference for a given domain $d$ is defined as

$$\Delta_{\mathrm{core}}^{(y_{i,j},d)} = \left| \mu^{(y_i,d)} - \mu^{(y_j,d)} \right|, \quad \forall y_i, y_j \in \mathcal{Y}, \ d \in \mathcal{D}, \ y_i \neq y_j.$$

The defined $\Delta_{\mathrm{spur}}^{(y,d_{i,j})} \in \mathbb{R}^p$ and $\Delta_{\mathrm{core}}^{(y_{i,j},d)} \in \mathbb{R}^p$ capture key structural variations in the data. To generalize these quantities across multiple classes and domains, we compute global directions using aggregated differences across groups. We define the **Spurious Direction** $\Delta_{\mathrm{spur}} \in \mathbb{R}^p$ as the average of differences in mean embeddings across domains within the same class, and the **Core Direction** $\Delta_{\mathrm{core}} \in \mathbb{R}^p$ as the average of differences across classes within the same domain

$$\Delta_{\mathrm{spur}} = \mathbb{E}_{y \in \mathcal{Y}} \left[ \Delta_{\mathrm{spur}}^{(y,d_{i,j})} \right], \quad \Delta_{\mathrm{core}} = \mathbb{E}_{d \in \mathcal{D}} \left[ \Delta_{\mathrm{core}}^{(y_{i,j},d)} \right],$$

To ensure proper interpretation based on theoretical insights, we normalize these direction vectors using the $\Sigma$-norm, defined as $\|v\|_\Sigma = \sqrt{v^\top \Sigma v}$ for $v \in \mathbb{R}^p$, where $\Sigma \in \mathbb{R}^{p \times p}$ denotes the empirical covariance matrix of the embedding vectors. This normalization accounts for the geometric structure of the embedding space, producing scalar quantities **Spurious Magnitude** $\|\Delta_{\mathrm{spur}}\|_\Sigma$ and **Core Magnitude** $\|\Delta_{\mathrm{core}}\|_\Sigma$ for use in regularization. We also derive regularization terms by analyzing how classifier weights align with the Spurious and Core Directions. The key idea is that overreliance on Spurious Directions leads to poor generalization on minority groups, while alignment with Core

Directions supports robust performance. Let $\beta^* = [\beta_1^*, \ldots, \beta_j^*, \ldots, \beta_m^*] \in \mathbb{R}^{p \times m}$ denote the current weight matrix of the classifier $f_\beta$, where each column $\beta_j^* \in \mathbb{R}^p$ corresponds to the weight vector for class $j$. During training, we use the current classifier weights to compute the alignment measures. To quantify the alignment between classifier weights and directional signals, we compute the average class-wise correlation with each direction using the $\Sigma$-norm. We refer to these measures as **Weight-Spurious Alignment** and **Weight-Core Alignment**

$$\text{cor}(\beta^*, \Delta_{\text{spur}}) = \frac{1}{m} \sum_{j=1}^m \frac{\langle \beta_j^*, \Delta_{\text{spur}} \rangle}{\|\beta_j^*\|_\Sigma \cdot \|\Delta_{\text{spur}}\|_\Sigma}, \quad \text{cor}(\beta^*, \Delta_{\text{core}}) = \frac{1}{m} \sum_{j=1}^m \frac{\langle \beta_j^*, \Delta_{\text{core}} \rangle}{\|\beta_j^*\|_\Sigma \cdot \|\Delta_{\text{core}}\|_\Sigma} \quad (1)$$

These correlation terms measure how closely the classifier's decision boundaries align with spurious or core feature directions, providing the basis for our embedding-level regularization.

Using this definition, we define the Spurious and Core Loss as:

$$\mathcal{L}_{\text{spur}} = \text{cor}(\beta^*, \Delta_{\text{spur}})\|\Delta_{\text{spur}}\|_\Sigma, \quad \mathcal{L}_{\text{core}} = \text{cor}(\beta^*, \Delta_{\text{core}})\|\Delta_{\text{core}}\|_\Sigma.$$

The **Spurious Loss** $\mathcal{L}_{\text{spur}}$ penalizes alignment with spurious directions, scaled by the magnitude of spurious variation in the data. The **Core Loss** $\mathcal{L}_{\text{core}}$ encourages alignment with core directions, with the negative sign ensuring that more substantial core alignment reduces the overall loss. Our loss formulation penalizes alignment with spurious directions and encourages alignment with core directions. By introducing control parameters $\lambda_{\text{spur}}$ and $\lambda_{\text{core}}$, we derive our embedding loss function:

$$\mathcal{L}_{\text{embedding}} = \lambda_{\text{spur}}\mathcal{L}_{\text{spur}} - \lambda_{\text{core}}\mathcal{L}_{\text{core}}. \quad (2)$$

The final SCER objective combines embedding loss with worst-group classification loss $\mathcal{L}_{\text{wge}}$:

$$\mathcal{L}_{\text{total}} = \mathcal{L}_{\text{wge}} + \mathcal{L}_{\text{embedding}}. \quad (3)$$

This loss formulation mitigates overfitting to domain-specific biases by reducing intra-class variation across domains while preserving inter-class separability. SCER promotes robust decision boundaries, enabling strong worst-group performance under distribution shifts. For additional implementation details, please refer to Appendix A.1.

### 3.3 THEORETICAL ANALYSIS OF SCER

We now present a theoretical analysis that formally decomposes worst-group error. The following theorem demonstrates that worst-group error can be decomposed into the classifier's reliance on spurious versus core components, thereby providing theoretical motivation for SCER's regularization of these components. Detailed proofs are provided in Appendix A.2.

**Theorem 1** (Worst-group Error Decomposition). *Consider a classification problem where each embedding vector $x \in \mathbb{R}^p$ is associated with a label $\mathcal{Y} = \{y_{-1}, y_{+1}\}$ and a domain $\mathcal{D} = \{d_R, d_G\}$. Each pair $(y, d) \in \mathcal{Y} \times \mathcal{D}$ defines a subpopulation. We assume that the data follows a group-conditional Gaussian distribution $x \mid (y, d) \sim \mathcal{N}(\mu^{(y,d)}, \Sigma)$ where $\Sigma$ is positive definite. We further assume that $\Delta_{core} = \mu^{(y_{+1},d)} - \mu^{(y_{-1},d)}$ is constant across $d$, $\Delta_{spur} = \mu^{(y,d_R)} - \mu^{(y,d_G)}$ is constant across $y$, $\Sigma^{(d_R)} = \Sigma^{(d_G)} = \Sigma$, $\mathbb{P}(y = \pm 1) = \mathbb{P}(d \in \{d_R, d_G\}) = \frac{1}{2}$, and $\mathbb{P}(y, d) \approx 0$ for some pairs $(y, d)$ under extreme spurious correlations. For classifier $f_\beta = sign(\beta^{*\top} x)$ with $\beta^* \in \mathbb{R}^p$, the worst-group error can be decomposed as*

$$E_{\text{wge}} = \Phi\left(\pm\frac{1}{2}\,\text{cor}(\beta^*, \Delta_{\text{spur}})\,\|\Delta_{\text{spur}}\|_\Sigma - \frac{1}{2}\,\text{cor}(\beta^*, \Delta_{\text{core}})\,\|\Delta_{\text{core}}\|_\Sigma\right).$$

**Remark** The sign $\pm$ in Theorem 1 reflects which subgroup constitutes the worst group. When $\Delta_{\text{spur}}^\top \beta^* > 0$, the unseen minority subgroups incur higher error, corresponding to the $+$ sign; the $-$ sign arises in the rare case where seen subgroups are worst. In the typical ERM regime with spurious correlations, the $+$ sign applies, and we retain $\pm$ for generality. See Appendix A.2 for the detailed derivation.

The Theorem shows that minimizing $E_{\text{wge}}$ requires reducing $\text{cor}(\beta^*, \Delta_{\text{spur}})\,\|\Delta_{\text{spur}}\|_\Sigma$ while simultaneously increasing $\text{cor}(\beta^*, \Delta_{\text{core}})\,\|\Delta_{\text{core}}\|_\Sigma$. The Weight-Spurious Alignment term $\text{cor}(\beta^*, \Delta_{\text{spur}})$

Table 1: Performance comparison on multiple datasets, each exhibiting different levels of spurious correlations. $^\dagger$ denotes the performance reported from Yang et al. (2023b) except for ColorMNIST. $^\ddagger$ denotes the performance reported from Deng et al. (2023) for Waterbird and CelebA. $^{\dagger\dagger}$ denotes the performance reported from Wen et al. (2025) for Waterbird and CelebA. For the remaining datasets, the detailed experimental settings follow Yang et al. (2023b), as described in Appendix B.1.2. SCER achieves the highest worst-group accuracy in Waterbirds, CelebA, MetaShift, and ColorMNIST, demonstrating its robustness in subpopulation shift scenarios.

| Algorithm | Waterbirds | | CelebA | | MetaShift | | ColorMNIST ($\rho = 80\%$) | |
|---|---|---|---|---|---|---|---|---|
| | Avg Acc | Worst Acc | Avg Acc | Worst Acc | Avg Acc | Worst Acc | Avg Acc | Worst Acc |
| ERM$^\dagger$ | 84.1 ±1.7 | 69.1 ±4.7 | 95.1 ±0.2 | 62.6 ±1.5 | 91.3 ±0.3 | 82.6 ±0.4 | 38.2 ±2.4 | 30.9 ±2.7 |
| GroupDRO$^\dagger$ | 88.8 ±1.8 | 78.6 ±1.0 | 91.4 ±0.6 | 89.0 ±0.7 | 91.0 ±0.1 | 85.6 ±0.4 | 73.5 ±0.3 | 73.1 ±0.1 |
| LISA$^\dagger$ | 92.8 ±0.2 | 88.7 ±0.6 | 92.6 ±0.1 | 86.3 ±1.2 | 89.5 ±0.4 | 84.1 ±0.4 | 73.8 ±0.1 | 73.2 ±0.1 |
| ReSample$^\dagger$ | 89.4 ±0.9 | 77.7 ±1.2 | 92.0 ±0.8 | 87.4 ±0.8 | 91.2 ±0.1 | 85.6 ±0.4 | 73.7 ±0.1 | 72.2 ±0.3 |
| PDE$^\ddagger$ | 92.4 ±0.8 | 90.3 ±0.3 | 92.4 ±0.8 | 91.0 ±0.4 | 87.4 ±0.1 | 78.1 ±0.1 | 76.6 ±0.8 | 72.9 ±0.3 |
| ElRep$^{\dagger\dagger}$ | 92.9 ±0.7 | 88.8 ±0.7 | 92.8 ±0.2 | 91.4 ±1.0 | 85.9 ±0.6 | 72.1 ±2.5 | 50.3 ±0.6 | 46.5 ±4.3 |
| SCER | 92.1 ±0.2 | **91.2 ±0.2** | 92.7 ±0.2 | **91.4 ±0.1** | 91.6 ±0.3 | **86.7 ±0.8** | 74.1 ±0.1 | **73.6 ±0.2** |

measures how much the classifier weights align with domain-specific variations within the same class, indicating the dependence on spurious features. The Spurious Magnitude $\|\Delta_{\mathrm{spur}}\|_\Sigma$ quantifies the divergence in the distribution of data between domains within the same label; thus, minimizing this component effectively reduces spurious correlations. In contrast, the Weight-Core Alignment term $\mathrm{cor}(\beta^*, \Delta_{\mathrm{core}})$ captures how well the classifier weights align with the true labels' discriminatory directions that are consistent across domains. The Core Magnitude $\|\Delta_{\mathrm{core}}\|_\Sigma$ captures the separation between different label distributions across domains, and augmenting this term enhances the model's ability to detect core predictive patterns.

## 4 EXPERIMENT

### 4.1 EXPERIMENT SETTING

**Datasets** We evaluate on both real-world and synthetic datasets. For vision tasks, we use Waterbirds (Sagawa et al.), CelebA (Liu et al., 2015), MetaShift (Liang & Zou, 2022), and ColorMNIST (Arjovsky et al., 2019). For language tasks, we use CivilComments (Borkan et al., 2019) and MultiNLI (Williams et al., 2018). These datasets are widely adopted benchmarks for evaluating robustness to spurious correlations. Detailed descriptions and statistics are in Appendix B.1.1.

**Models** Following prior work (Gulrajani & Lopez-Paz; Izmailov et al., 2022), we use pretrained ResNet-50 (He et al., 2016) for image datasets and pretrained BERT (Idrissi et al., 2022) for text datasets. We use SGD with momentum (Polyak, 1964) for images and AdamW (Loshchilov & Hutter, 2017) for text, following standard protocols (Yang et al., 2023b). Training steps are 5,000 for Waterbirds, MetaShift, and ColorMNIST, and 30,000 for CelebA, CivilComments, and MultiNLI.

**Baselines** We evaluate our method against various bias mitigation and imbalanced learning algorithms. Our comparison includes ERM; subpopulation robustness methods such as GroupDRO (Sagawa et al.), LISA (Yao et al., 2022), PDE (Deng et al., 2023), and ElRep (Wen et al., 2025); and class imbalance techniques such as ReSample (Japkowicz, 2000). For ColorMNIST, we evaluate only methods that achieve worst-group accuracy above 70%. We conduct experiments on a broader set of baselines, with detailed descriptions and complete results provided in Appendix B.1.2 and Tables 11, 12, 13, and 15. We also conduct a focused comparison against GroupDRO-ES (Izmailov et al., 2022), the most competitive GroupDRO variant with early stopping, as presented in Table 16.

**Evaluation Metrics** Following prior work, we use Worst-group Accuracy (Worst Acc) as the primary metric to assess performance on the most vulnerable minority group, and Average Accuracy (Avg Acc) for overall performance. Our objective is to minimize the gap between these metrics.

## 5 RESULTS

### 5.1 QUANTITATIVE ANALYSIS

**Image Dataset** Table 1 shows that SCER outperforms baseline methods across four benchmarks: Waterbirds, CelebA, MetaShift, and ColorMNIST, each presenting different spurious correlation

Table 2: Performance comparison on ColorMNIST under increasing spurious correlation levels. SCER achieves the highest worst-group accuracy across different spurious correlation levels, demonstrating robustness to spurious feature reliance in ColorMNIST.

| | ColorMNIST | | | | | | | |
|---|---|---|---|---|---|---|---|---|
| | $\rho = 80\%$ | | $\rho = 90\%$ | | $\rho = 95\%$ | | $\rho = 99\%$ | |
| Algorithm | Avg Acc | Worst Acc | Avg Acc | Worst Acc | Avg Acc | Worst Acc | Avg Acc | Worst Acc |
| ERM | 38.2 ±2.4 | 30.9 ±2.7 | 31.9 ±7.4 | 7.8 ±3.4 | 41.1 ±13.0 | 10.1 ±4.5 | 50.7 ±8.2 | 8.5 ±5.0 |
| GroupDRO | 73.5 ±0.3 | 73.1 ±0.1 | 73.5 ±0.0 | 72.7 ±0.3 | 72.3 ± 0.1 | 70.7 ±0.2 | 50.3 ±3.9 | 38.7 ±0.8 |
| LISA | 73.8 ±0.1 | 73.2 ±0.1 | 73.3 ±0.1 | 72.9 ±0.2 | 72.6 ±0.4 | 71.4 ±0.6 | 53.1 ±7.2 | 10.2 ±8.1 |
| ReSample | 73.7 ±0.1 | 72.2 ±0.3 | 72.6 ±0.5 | 70.8 ±1.6 | 71.8 ±0.1 | 70.7 ±0.5 | 57.1 ±3.0 | 45.4 ±10.0 |
| PDE | 76.6 ±0.8 | 72.9 ±0.3 | 72.6 ±0.2 | 70.2 ±0.7 | 73.4 ±0.4 | 70.0 ±0.3 | 56.3 ±4.3 | 47.5 ±8.9 |
| SCER | 74.1 ±0.1 | **73.6 ±0.2** | 73.6 ±0.1 | **73.0 ±0.1** | 73.5 ±0.3 | **72.8 ±0.3** | 61.0 ±2.0 | **56.0 ±2.2** |

challenges. SCER achieves the highest worst-group accuracy in four datasets, with 91.2% on Waterbirds, 91.4% on CelebA, 86.7% on MetaShift, and 73.6% on ColorMNIST at $\rho = 80\%$. These results demonstrate that embedding-level disentanglement effectively mitigates spurious bias.

**Image Dataset with Stronger Spurious Correlations**    Table 2 reports performance on ColorMNIST under varying levels of spurious correlation $\rho$. As $\rho$ increases, the dataset becomes increasingly biased, making classification more challenging and highlighting the detrimental impact of spurious correlations on standard models. SCER consistently achieves the highest worst-group accuracy across all values of $\rho$, attaining 73.6% at $\rho = 80\%$, 73.0% at $\rho = 90\%$, 72.8% at $\rho = 95\%$, and 56.0% at $\rho = 99\%$, outperforming all baselines.

Table 3: Performance on ColorMNIST with one group absent. SCER achieves the highest worst-group accuracy.

| ColorMNIST (One Subpopulation Omitted) | | |
|---|---|---|
| Algorithm | Avg Acc | Worst Acc |
| ERM | 52.9 ±8.4 | 9.7 ±5.4 |
| GroupDRO | 53.6 ±2.5 | 44.1 ±6.2 |
| LISA | 49.5 ±2.4 | 13.5 ±11.0 |
| ReSample | 49.1 ±0.8 | 16.3 ±13.0 |
| PDE | 71.0 ±12.6 | 8.3 ±13.9 |
| SCER | 65.3 ±1.0 | **59.6 ±1.0** |

In addition to the standard evaluation settings in Table 2, we examine a more challenging scenario in which a minority group is completely absent during training, creating a label color distribution of 50-0-10-40, as shown in Table 8. This extreme setting tests algorithms' ability to generalize under harsh spurious correlations when facing complete group absence. As shown in Table 3, SCER achieves 59.6% worst-group accuracy in this extreme setting, significantly outperforming all existing methods. This result reveals limitations of existing approaches under extreme spurious correlations.

In this extreme setting where an entire subpopulation is omitted from training, GroupDRO (Sagawa et al.) suffers from instability with extreme minority groups, poor extrapolation to unseen subpopulations, and regularization-capacity trade-offs that limit its effectiveness. Class imbalance methods (Japkowicz, 2000) and LISA (Yao et al., 2022) fail as they rely on reweighting available samples or interpolating between seen domains, making them unable to handle completely missing groups. Similarly, PDE (Deng et al., 2023), despite initially learning from balanced data, cannot generalize to entirely unseen group combinations. These results show embedding-level interventions are essential when complete groups are absent, as this causes extreme spurious correlations. SCER's superior performance stems from directly regularizing the representation space, mitigating spurious correlations and enabling robust generalization to unseen subpopulations.

Table 4: Performance comparison without explicit bias labels.

| ColorMNIST (Two train envs) | |
|---|---|
| Method | Avg Acc |
| IRM | 54.8 ±8.3 |
| EIIL (IRM) | 58.5 ±1.8 |
| EIIL + DRO | 68.2 ±7.0 |
| EIIL + SCER | **72.6 ±5.5** |

**Integration with Environment Inference Methods.**    A key advantage of SCER is its modular design, which enables seamless integration with existing frameworks without requiring explicit bias labels. We integrate SCER with the Environment Inference for Invariant Learning (EIIL) framework (Creager et al., 2021), replacing EIIL's second-stage IRM objective with SCER. Table 4 shows experimental results integrating EIIL with SCER where no environment information is provided, simulating realistic scenarios without environment labels.

We adopt the data setup from (Creager et al., 2021), with two environments differing in the strength of spurious correlations, as detailed in Table 8. The environments inferred by EIIL achieve over 95% agreement with actual spurious groups in ColorMNIST, providing reliable pseudo-labels for SCER training. As shown in Table 4, SCER maintains its effectiveness even with inferred environments, achieving the highest test accuracy of 72.6%. Notably, while GroupDRO suffers significant performance degradation under environmental misalignment, SCER remains robust due to its embedding-level regularization. This demonstrates SCER's practical applicability and adaptability to real-world scenarios where explicit bias annotations are unavailable. Even in scenarios where environment labels are increasingly noisy, SCER continues to demonstrate robust performance, as detailed in Table 14.

**Text Dataset**   Table 5 presents SCER's performance on text-based datasets. SCER consistently achieves the best results on both CivilComments for multi-domain evaluation and MultiNLI for multi-class evaluation.

On CivilComments, SCER achieves the highest worst-group accuracy of 74.0%, effectively mitigating performance degradation in minority subpopulations. On MultiNLI, SCER achieves the highest worst-group accuracy of 76.8%, demonstrating balanced learning across different class distributions. These results confirm that SCER's embedding regularization approach generalizes effectively beyond vision to text-based scenarios.

Table 5: Performance comparison on text datasets. [‡] from Deng et al. (2023) for Civilcomments. [††] from Wen et al. (2025) for Civilcomments. SCER achieves the best worst-group accuracy across multi-class and domain settings.

| Algorithm | CivilComments | | MultiNLI | |
|---|---|---|---|---|
| | Avg Acc | Worst Acc | Avg Acc | Worst Acc |
| ERM[†] | 85.4 ±0.2 | 63.7 ±1.1 | 80.9 ±0.1 | 66.8 ±0.5 |
| GroupDRO[†] | 81.8 ±0.6 | 70.6 ±1.2 | 81.1 ±0.3 | 76.0 ±0.7 |
| LISA[†] | 82.7 ±0.1 | 73.7 ±0.3 | 80.3 ±0.4 | 73.3 ±1.0 |
| ReSample[†] | 82.2 ±0.0 | 73.3 ±0.5 | 77.2 ±0.2 | 72.3 ±0.8 |
| PDE[‡] | 86.3 ±1.7 | 71.5 ±0.5 | 69.1 ±0.3 | 65.8 ±0.6 |
| ElRep[††] | 79.0 ±0.7 | 70.5 ±0.5 | 69.0 ±3.8 | 66.8 ±5.2 |
| SCER | 81.7 ±0.4 | **74.0 ±1.0** | 80.4 ±0.1 | **76.8 ±0.5** |

SCER demonstrates robust performance across multi-domain and multi-class environments, highlighting its versatility in handling subpopulation shifts across modalities.

Our quantitative analysis shows that SCER consistently surpasses existing methods across diverse image and text settings, maintaining strong worst-group performance even under severe spurious correlations, missing groups, and inferred environments. This demonstrates that embedding-level regularization effectively reduces spurious bias and supports robust generalization across domains.

## 5.2   QUALITATIVE ANALYSIS

Table 6: Sensitivity analysis of embedding regularization on ColorMNIST. Worst-group accuracy remains stable across settings, with the best results observed when both loss terms are combined.

| ColorMNIST ($\rho = 95\%$) | | | | | | | | |
|---|---|---|---|---|---|---|---|---|
| both $\lambda = 0$ | | | $\lambda_{core}$ (fixed $\lambda_{spur}$) | | | $\lambda_{spur}$ (fixed $\lambda_{core}$) | | |
| Setting | Avg Acc | Worst Acc | $\lambda_{core}$ | Avg Acc | Worst Acc | $\lambda_{spur}$ | Avg Acc | Worst Acc |
| 0 | 72.3 ±0.1 | 70.7 ±0.2 | 0.0 | 72.8 ±0.4 | 71.6 ±0.6 | 0.0 | 72.8 ±0.3 | 71.6 ±0.4 |
| – | – | – | 0.5 | 73.0 ±0.2 | **72.2 ±0.4** | 0.5 | 72.3 ±0.4 | 71.6 ±0.3 |
| – | – | – | 1.0 | 73.3 ±0.2 | 72.0 ±0.2 | 1.0 | 73.5 ±0.3 | **72.8 ±0.3** |

**Embedding Regularization Analysis**   We conduct sensitivity analysis on ColorMNIST with $\rho = 95\%$ to evaluate our proposed loss components. As shown in Table 6, we systematically vary $\lambda_{core}$ and $\lambda_{spur}$ according to Equation 2. Results show that each regularization component independently improves worst-group accuracy over the baseline, demonstrating that both $\mathcal{L}_{core}$ and $\mathcal{L}_{spur}$ contribute to robustness against spurious correlations. Notably, their joint optimization achieves the highest worst-group performance, confirming that the two loss terms produce complementary effects. This validates our theoretical insight that achieving optimal worst-group generalization requires simultaneously maximizing alignment with core features while minimizing reliance on spu-

rious features. To further assess parameter sensitivity in a large-scale setting, we provide additional analysis on CelebA in Table 18.

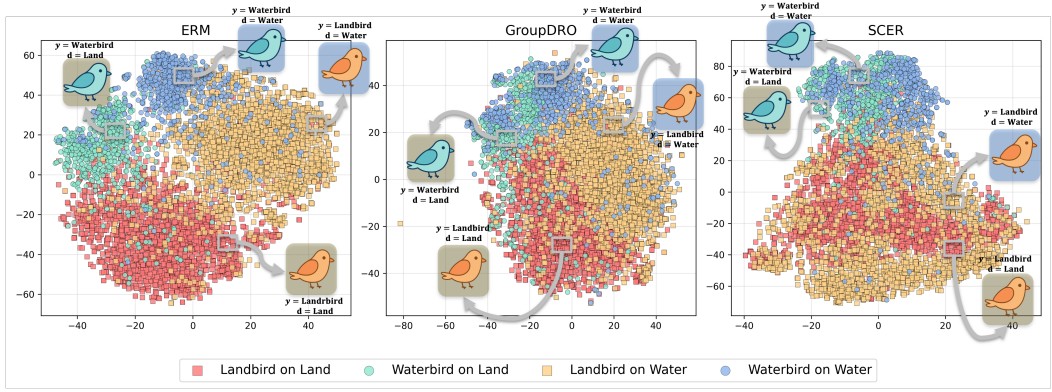

Figure 3: t-SNE visualization on the Waterbirds dataset. ERM clusters samples primarily by background rather than label. GroupDRO partially mitigates this but still exhibits background-based separation within each label. SCER produces label-aligned, background-invariant embeddings, effectively suppressing spurious correlations.

**Embedding Space Analysis**    Figure 3 shows t-SNE projections of embeddings learned by the three models on the Waterbirds dataset. ERM exhibits strong domain-based clustering, with samples separating predominantly by background rather than label. Samples from different backgrounds form distinct clusters, indicating heavy reliance on spurious features rather than core class characteristics. GroupDRO shows reduced but still visible clustering by background within the same label, demonstrating incomplete suppression of spurious correlations.

In contrast, SCER produces label-aligned embeddings where samples cluster primarily by label regardless of background. Waterbird samples from different backgrounds intermix to form a single unified cluster, while Landbird samples from different backgrounds aggregate similarly, maintaining clear separation between the two classes. This domain-invariant structure provides qualitative evidence that SCER effectively suppresses spurious correlations via embedding-level regularization, resulting in representations that generalize robustly across distribution shifts.

Table 7: Comparison between Euclidean norm and $\Sigma$-norm on ColorMNIST. Our proposed $\Sigma$-norm consistently achieves higher worst-group accuracy.

| **ColorMNIST ($\rho = 95\%$)** | | |
|---|---|---|
| Norm Type | Avg Acc | Worst Acc |
| Euclidean norm | 72.7 $\pm$0.8 | 70.0 $\pm$0.4 |
| $\Sigma$-norm (Ours) | 73.5 $\pm$0.3 | **72.8 $\pm$0.3** |

**Component Analysis**    We investigate our proposed $\Sigma$-norm by replacing the standard Euclidean norm in Equation 1. As shown in Table 7, the $\Sigma$-norm outperforms the Euclidean baseline. This improvement comes from the natural benefits of the distance, which captures the structure of covariance of the features and ensures the invariance of the scale in high-dimensional spaces (Ghorbani, 2019; Mahalanobis, 2018).

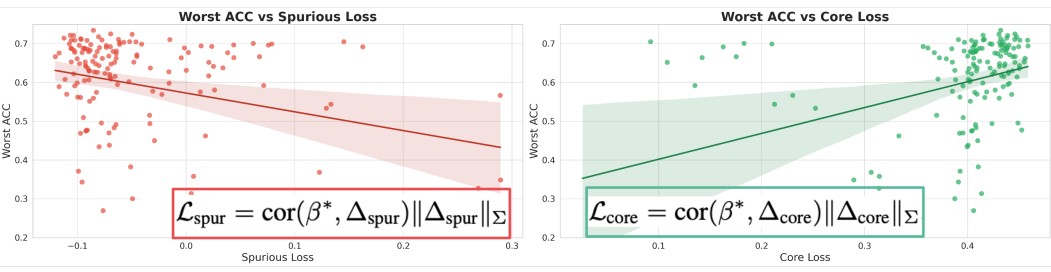

Figure 4: Scatter plots showing worst-group accuracy vs spurious and core metrics. Spurious Loss shows negative correlation, while Core Loss shows positive correlation.

**Correlation Analysis between Objective Function Decomposition and worst-group accuracy**
Figure 4 demonstrates the relationship between worst-group accuracy and spurious/core metrics, revealing insights into group-invariant learning. Spurious Loss exhibits negative correlations with Worst Acc, indicating that a stronger dependence of the classifier on domain-specific characteristics degrades the performance of the minority group, while Core Loss shows positive correlations, demonstrating that alignment with genuine label discriminatory features enhances the generalization of the group.

Specifically, Weight-Spurious Alignment $\text{cor}(\beta^*, \Delta_{\text{spur}})$ measures how much classifier weights align with intra-class domain variations, quantifying spurious feature dependence. In contrast, Weight-Core Alignment $\text{cor}(\beta^*, \Delta_{\text{core}})$ captures how well the classifier aligns with domain-consistent discriminative directions. Spurious Magnitude $\|\Delta_{\text{spur}}\|_{\Sigma}$ and Core Magnitude $\|\Delta_{\text{core}}\|_{\Sigma}$ quantify inter-domain distributional differences and inter-label separability, respectively. The consistent patterns observed across these metrics align with our theoretical decomposition, confirming that the regularization terms directly address the properties essential for worst-group robustness. These results establish that balancing core signal capture and spurious correlation suppression is fundamental to group invariance.

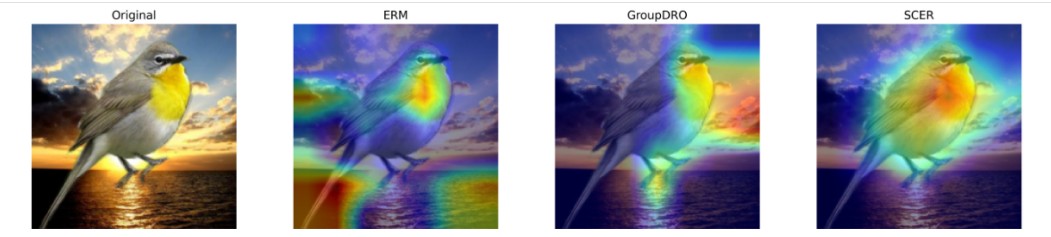

Figure 5: Grad-CAM visualization comparison on the Waterbirds. ERM heavily relies on background cues, and GroupDRO only partially captures the object, whereas SCER consistently attends to bird-specific features. This demonstrates SCER's ability to suppress spurious background correlations while focusing on label-relevant structures.

**Grad-CAM** Figure 5 presents a comparative analysis of Grad-CAM (Selvaraju et al., 2017) visualizations for three methods applied to bird classification models. ERM exhibits dispersed activation patterns across the entire image, allocating considerable attention to background regions. GroupDRO shows concentrated activation on some regions of the bird, but does not fully utilize all parts that can be considered core features for classification. In contrast, SCER concentrates strong activation across structural features such as the bird's body, head, wings, and tail, while effectively suppressing dependence on background or spurious elements. These results visually demonstrate that our mean embedding-based penalty approach comprehensively captures diverse discriminative features critical for classification while effectively suppressing spurious factors. Additional examples are provided in Figure 6.

Our qualitative analysis demonstrates that SCER effectively suppresses spurious features and learns representations focused on core signals, as evidenced by embedding structures, attention visualizations, and interpretability analyses that support robust group-invariant behavior.

## 6 CONCLUSION

We propose Spurious Correlation-Aware Embedding Regularization (SCER), which addresses spurious correlations by decomposing worst-group error into spurious and core components at the embedding level. By suppressing alignment with spurious feature directions while promoting consistency with core feature directions, the model is prevented from overfitting to domain-specific artifacts. Through extensive experiments across image and text domains, we confirm that SCER consistently achieves superior worst-group accuracy, demonstrating that explicit embedding-level regularization addresses subgroup bias more directly than sample reweighting or feature alignment approaches. Furthermore, SCER's modular design enables integration with environment-inference methods, making it practically applicable to real-world scenarios without requiring explicit bias labels. This work presents a promising direction for robust learning under distribution shift and is expected to contribute to practical applications where fairness is critical.

ACKNOWLEDGMENTS

This work was supported by the National Research Foundation of Korea(NRF) grant funded by the Korea government(MSIT)(RS-2024-00457216).

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

# A    METHODOLOGY

This section provides comprehensive algorithmic details and theoretical proofs for the SCER framework presented in the main paper. We first present detailed step-by-step procedures for SCER implementation, including covariance computation, embedding regularization, and loss optimization, as summarized in the main text. Subsequently, we provide formal proofs for the theoretical foundations underlying our embedding regularization approach, including the decomposition of the embedding space into core and spurious components and the optimality conditions for robustness.

## A.1    OVERALL PROCESS OF SCER

Recall from Section 3 that we define the Spurious and Core Loss as:

$$\mathcal{L}_{\text{spur}} = \text{cor}(\beta^*, \Delta_{\text{spur}})\|\Delta_{\text{spur}}\|_\Sigma, \quad \mathcal{L}_{\text{core}} = \text{cor}(\beta^*, \Delta_{\text{core}})\|\Delta_{\text{core}}\|_\Sigma.$$

The **Spurious Loss** $\mathcal{L}_{\text{spur}}$ penalizes alignment with spurious directions, scaled by the magnitude of spurious variation in the data. The **Core Loss** $\mathcal{L}_{\text{core}}$ encourages alignment with core directions, with the negative sign ensuring that more substantial core alignment reduces the overall loss. Our loss formulation penalizes alignment with spurious directions and encourages alignment with core directions. By introducing control parameters $\lambda_{\text{spur}}$ and $\lambda_{\text{core}}$, we derive our embedding loss function:

$$\mathcal{L}_{\text{embedding}} = \lambda_{\text{spur}}\mathcal{L}_{\text{spur}} - \lambda_{\text{core}}\mathcal{L}_{\text{core}}.$$

The final SCER objective combines embedding loss with worst-group classification loss $\mathcal{L}_{\text{wge}}$:

$$\mathcal{L}_{\text{total}} = \mathcal{L}_{\text{wge}} + \mathcal{L}_{\text{embedding}}.$$

This loss formulation mitigates overfitting to domain-specific biases by reducing intra-class variation across domains while preserving inter-class separability. SCER promotes robust decision boundaries, enabling strong worst-group performance under distribution shifts.

As shown in Equation 3, the total training loss combines two complementary components. The first term, $\mathcal{L}_{\text{wge}}$, is the worst-group classification loss derived from GroupDRO, which addresses subpopulation imbalances through robust optimization (**Worst-Group Error-based Classification**). The second term, $\mathcal{L}_{\text{embedding}}$, is an embedding-level regularization that explicitly controls the alignment between classifier weights and Spurious/Core Directions in the feature space (**Embedding-based Regularization**). This regularization penalizes the classifier's reliance on domain-specific artifacts while encouraging alignment with label-consistent features. The two components work synergistically: the classification loss ensures robust performance across subpopulations, while the embedding regularization structures the representation space for better generalization. The combined objective is optimized via gradient-based methods (**Loss Optimization**). The details of each step are illustrated in Algorithm 1.

---

**Algorithm 1** SCER Training Process

**Require:** Data $(x, y, d)$, parameters $(w, \beta)$, hyperparams $(\eta, \lambda_{\text{spur}}, \lambda_{\text{core}})$
    Initialize $\mathbf{q} \leftarrow \mathbf{1}, \Sigma \leftarrow \mathbf{I}$
    **for** each step **do**
        $x_{emb} \leftarrow f_w(x)$
        **(1) Worst-Group Error based Classification**
        $\mathcal{L}(y, f_\beta(x_{\text{emb}}))$,
        $q_{(y,d)} \leftarrow q_{(y,d)} \exp(\eta \mathbb{E}[\mathcal{L}_{(y,d)}])$
        $q \leftarrow q / \sum_{(y,d) \in \mathcal{Y} \times \mathcal{D}} q_{(y,d)}$
        $\mathcal{L}_{\text{wge}} = \sum_{(y,d) \in \mathcal{Y} \times \mathcal{D}} q_{(y,d)} \mathbb{E}[\mathcal{L}_{(y,d)}]$
        **(2) Embedding-based Regularization**
        $\Sigma = \frac{1}{N}\sum_i (x_{emb,i} - \bar{x}_{emb})(x_{emb,i} - \bar{x}_{emb})^T$
        $\beta^\star \leftarrow$ current classifier weights
        $\text{cor}(\beta^*, \Delta_{\text{spur}}) = \frac{1}{m}\sum_{j=1}^m \frac{\langle \beta_j^*, \Delta_{\text{spur}} \rangle}{\|\beta_j^*\|_\Sigma \cdot \|\Delta_{\text{spur}}\|_\Sigma}$
        $\text{cor}(\beta^*, \Delta_{\text{core}}) = \frac{1}{m}\sum_{j=1}^m \frac{\langle \beta_j^*, \Delta_{\text{core}} \rangle}{\|\beta_j^*\|_\Sigma \cdot \|\Delta_{\text{core}}\|_\Sigma}$
        $\mathcal{L}_{\text{spur}} = \text{cor}(\beta^*, \Delta_{\text{spur}})\|\Delta_{\text{spur}}\|_\Sigma$
        $\mathcal{L}_{\text{core}} = \text{cor}(\beta^*, \Delta_{\text{core}})\|\Delta_{\text{core}}\|_\Sigma$
        $\mathcal{L}_{\text{embedding}} = \lambda_{\text{spur}}\mathcal{L}_{\text{spur}} - \lambda_{\text{core}}\mathcal{L}_{\text{core}}$
        **(3) Optimize**
        $\mathcal{L}_{\text{total}} = \mathcal{L}_{\text{wge}} + \mathcal{L}_{\text{embedding}}$
        Update $(w, \beta)$ via gradient descent
    **end for**

---

**Worst-Group Error-based Classification.**    The feature vector $x_{\text{emb}} \in \mathbb{R}^p$, where $p$ is the embedding dimension, is used to compute classification losses for each group by comparing predictions

with ground-truth labels. These group-wise losses are aggregated, and group-specific weights are updated using an exponentiated gradient scheme. This reweighting emphasizes underperforming groups, helping to correct distributional imbalances and enhancing robustness to worst-group errors.

**Embedding-based Regularization.** This step regularizes the learned representation by explicitly distinguishing spurious and core directions in the embedding space and aligning the decision boundary accordingly. To capture group-wise variability, a covariance matrix $\Sigma \in \mathbb{R}^{p \times p}$ is computed from the feature embeddings $x_{\text{emb}}$. The core and spurious directions, denoted by $\Delta_{\text{core}} \in \mathbb{R}^p$ and $\Delta_{\text{spur}} \in \mathbb{R}^p$ respectively, are derived from the groupwise mean embeddings. To assess the classifier's alignment with these directions, we use the current classifier weights $\beta_j^* \in \mathbb{R}^p$ for each class $j$, which are dynamically updated during training via gradient descent. Using the weight matrix $\beta^*$, the spurious and core components—$\mathcal{L}_{\text{spur}}$ and $\mathcal{L}_{\text{core}}$—are computed. Their weighted combination forms the directional loss $\mathcal{L}_{\text{embedding}}$. Minimizing this objective reduces the classifier's reliance on spurious directions while encouraging alignment with core directions, guiding the representation toward meaningful semantic structure. This regularization promotes intra-class alignment across domains and enhances inter-class separation, thereby mitigating spurious correlations in the embedding space and reducing worst-group error.

**Loss Optimization.** The total loss is computed by combining the worst-group-error loss $\mathcal{L}_{\text{wge}}$ with the directional regularization loss $\mathcal{L}_{\text{embedding}}$. Minimizing this objective ensures that the classifier performs well across all groups while learning representations that are robust to spurious correlations. By aligning the embedding space with core directions and suppressing spurious ones, the method enhances robustness to worst-group errors under distribution shift.

## A.2 THEORETICAL ANALYSIS OF SCER

We now present a theoretical analysis that formally decomposes the worst-group error. Following prior work (Yao et al., 2022), we adopt a Gaussian mixture model to analyze how worst-group error arises from label-domain interactions.

In the main text, we decompose the classifier $f$ into a feature extractor $f_w$ and a linear classifier $f_\beta$, where the representation is given by the embedding $x_{\text{emb}} = f_w(x) \in \mathbb{R}^p$. Throughout this subsection, we conduct the analysis *in the embedding space* and, with a slight abuse of notation, we overload the symbol $x$ to denote the embedding:

$$x \equiv x_{\text{emb}}$$

Consequently, all distributions of the form $x \mid (y, d)$ below are to be understood as distributions over embedding vectors.

### A.2.1 BINARY SETTING

Consider a classification problem where each embedding vector $x \in \mathbb{R}^p$ is associated with a label $\mathcal{Y} = \{y_{-1}, y_{+1}\}$ and a domain $\mathcal{D} = \{d_R, d_G\}$. Each pair $(y, d) \in \mathcal{Y} \times \mathcal{D}$ defines a *subpopulation*. We assume that the data follows a group-conditional Gaussian distribution $x \mid (y, d) \sim \mathcal{N}(\mu^{(y,d)}, \Sigma)$ where $\Sigma$ is positive definite. We further assume that $\Delta_{\text{core}} = \mu^{(y_{+1},d)} - \mu^{(y_{-1},d)}$ is constant across $d$, $\Delta_{\text{spur}} = \mu^{(y,d_R)} - \mu^{(y,d_G)}$ is constant across $y$, $\Sigma^{(d_R)} = \Sigma^{(d_G)} = \Sigma$, $\mathbb{P}(y = \pm 1) = \mathbb{P}(d \in \{d_R, d_G\}) = \frac{1}{2}$, and $\mathbb{P}_{y,d} \approx 0$ for some pairs $(y, d)$ under extreme spurious correlations.

**Proposition 1** (ERM solution under Cross-Entropy Loss). *Under this assumption, the ERM solution with cross-entropy loss satisfies*

$$\beta^* \propto \Sigma^{-1} \tilde{\Delta}, \qquad \tilde{\Delta} = \Delta_{\text{core}} + \Delta_{\text{spur}},$$

*Proof.* The ERM problem using cross-entropy loss as a logistic regression surrogate is

$$\min_{\beta, \beta_0} \mathbb{E}\left[\log\left(1 + \exp(-y(\beta^\top x + \beta_0))\right)\right].$$

The population minimizer $\beta^*$ satisfies the score equation

$$\mathbb{E}\big[y\, x\, \sigma(-y(\beta^{*\top}x + \beta_0^*))\big] = 0, \qquad \sigma(t) = \frac{1}{1+e^{-t}}.$$

Under the Gaussian assumption with shared covariance $\Sigma$, the Bayes classifier is derived from the log-likelihood ratio

$$\log \frac{p(x \mid y_{+1})}{p(x \mid y_{-1})} = (\mu^{(y_{+1})} - \mu^{(y_{-1})})^\top \Sigma^{-1} x - \tfrac{1}{2}(\mu^{(y_{+1})} + \mu^{(y_{-1})})^\top \Sigma^{-1}(\mu^{(y_{+1})} - \mu^{(y_{-1})}),$$

which is linear in $x$ with a normal vector $\Sigma^{-1}(\mu^{(y_{+1})} - \mu^{(y_{-1})})$. Since logistic regression is Fisher-consistent under correct model specification, the cross-entropy minimizer $\beta^*$ aligns with the Bayes direction:

$$\beta^* \propto \Sigma^{-1}(\mu^{(y_{+1})} - \mu^{(y_{-1})}).$$

In the extreme shift setting, the effective class means correspond to $\mu^{(y_{+1},d_R)}$ and $\mu^{(y_{-1},d_G)}$. Thus

$$\beta^* \propto \Sigma^{-1}\big(\mu^{(y_{+1},d_R)} - \mu^{(y_{-1},d_G)}\big).$$

Decomposing the difference:

$$\mu^{(y_{+1},d_R)} - \mu^{(y_{-1},d_G)} = \big(\mu^{(y_{+1},d_R)} - \mu^{(y_{-1},d_R)}\big) + \big(\mu^{(y_{-1},d_R)} - \mu^{(y_{-1},d_G)}\big) = \Delta_{\text{core}} + \Delta_{\text{spur}} = \tilde{\Delta},$$

we conclude

$$\beta^* \propto \Sigma^{-1}\tilde{\Delta}.$$

$\square$

**Remark** Our theoretical analysis relies on modeling the embedding distribution $x \in \mathbb{R}^p$ under each subpopulation $(y, d)$ using a Gaussian with a shared covariance matrix $\Sigma$. This assumption enables the tractable LDA-style derivation of Proposition 1 and Theorem 1, where the classifier direction admits the closed-form expression $\beta^* \propto \Sigma^{-1}\tilde{\Delta}$. Importantly, this assumption is empirically well-grounded. Seddik et al. (2020) show that deep embeddings exhibit a dominant shared covariance structure across groups, with group-specific differences appearing as higher-order perturbations. Furthermore, the neural collapse phenomenon indicates that within-class covariances converge toward one another as training progresses (Papyan et al., 2020).

**Theorem 1** (Worst-group error Decomposition). *Under this assumption, for classifier $f_\beta = \text{sign}(\beta^{*\top}x)$ with $\beta^* \in \mathbb{R}^p$, the worst-group error can be decomposed as*

$$E_{\text{wge}} = \Phi\big(\pm\tfrac{1}{2}\,\text{cor}(\beta^*, \Delta_{\text{spur}})\,\|\Delta_{\text{spur}}\|_\Sigma - \tfrac{1}{2}\,\text{cor}(\beta^*, \Delta_{\text{core}})\,\|\Delta_{\text{core}}\|_\Sigma\big),$$

*Proof.* Consider the linear classifier

$$f(x) = \text{sign}(\beta^\top x + \beta_0).$$

For subgroup $(y, d)$,

$$\mathbb{P}(f(x) \neq y \mid (y, d)) = \Phi\left(-\frac{y(\beta^\top \mu^{(y,d)} + \beta_0)}{\|\beta\|_\Sigma}\right), \qquad \|\beta\|_\Sigma = \sqrt{\beta^\top \Sigma \beta}.$$

Using $\beta_0 = -\beta^\top \mathbb{E}[x]$ under extreme spurious correlations, we obtain the following.

Under the extreme shift, the effective training distribution consists of $(y_{+1}, d_R)$ and $(y_{-1}, d_G)$, so $\mathbb{E}[x] = \tfrac{1}{2}(\mu^{(y_{+1},d_R)} + \mu^{(y_{-1},d_G)})$ and $\beta_0 = -\beta^\top \mathbb{E}[x]$. Substituting into the subgroup error formula gives $\mathbb{P}(f(x) \neq y \mid (y, d)) = \Phi\big(-\frac{y\beta^\top(\mu^{(y,d)} - \mathbb{E}[x])}{\|\beta\|_\Sigma}\big)$. For $(y_{+1}, d_R)$: $\mu^{(y_{+1},d_R)} - \mathbb{E}[x] = \tfrac{1}{2}(\mu^{(y_{+1},d_R)} - \mu^{(y_{-1},d_G)}) = \tfrac{1}{2}\tilde{\Delta}$, hence $E(y_{+1}, d_R) = \Phi\big(-\frac{\tilde{\Delta}^\top \beta}{2\|\beta\|_\Sigma}\big)$. Applying the same procedure to all four subgroups:

$$E(y_{+1}, d_R) = \Phi\left(-\frac{1}{2}\frac{\tilde{\Delta}^\top \beta}{\|\beta\|_\Sigma}\right), \qquad \text{(seen)}$$

$$E(y_{-1}, d_R) = \Phi\left(\frac{(\frac{1}{2}\tilde{\Delta}-\Delta_{\text{core}})^\top \beta}{\|\beta\|_\Sigma}\right), \qquad \text{(unseen)}$$

$$E(y_{+1}, d_G) = \Phi\left(\frac{(\frac{1}{2}\tilde{\Delta}-\Delta_{\text{core}})^\top \beta}{\|\beta\|_\Sigma}\right), \qquad \text{(unseen)}$$

$$E(y_{-1}, d_G) = \Phi\left(-\frac{1}{2}\frac{\tilde{\Delta}^\top \beta}{\|\beta\|_\Sigma}\right). \qquad \text{(seen)}$$

Let

$$z = \frac{1}{2}\frac{\tilde{\Delta}^\top \beta}{\|\beta\|_\Sigma}.$$

Then

$$E(y_{+1}, d_R) = \Phi(-z), \qquad E(y_{-1}, d_G) = \Phi(-z).$$

By Proposition 1, $\beta^* \propto \Sigma^{-1}\tilde{\Delta}$, so $\tilde{\Delta}^\top \beta^* \propto \|\tilde{\Delta}\|^2_{\Sigma^{-1}} > 0$, implying $z > 0$. Both seen subgroups therefore have identical error $\Phi(-z) < \Phi(0) = 0.5$. The unseen subgroups $(y_{-1}, d_R), (y_{+1}, d_G)$ have error $\Phi(w)$ where $w = \frac{(\frac{1}{2}\tilde{\Delta}-\Delta_{\text{core}})^\top \beta}{\|\beta\|_\Sigma}$. Note that $w = z - \frac{\Delta_{\text{core}}^\top \beta}{\|\beta\|_\Sigma}$, so increasing the core alignment $\Delta_{\text{core}}^\top \beta$ reduces $w$ and thus $\Phi(w)$.

$$E_{\text{wge}} = \max\{\Phi(-z), \Phi(w)\}.$$

Finally, writing $\tilde{\Delta} = \Delta_{\text{spur}} + \Delta_{\text{core}}$ and substituting into $w$ and $-z$:

$$w = \frac{1}{2}\frac{(\Delta_{\text{spur}}+\Delta_{\text{core}}-2\Delta_{\text{core}})^\top \beta}{\|\beta\|_\Sigma} = +\frac{1}{2}\frac{\Delta_{\text{spur}}^\top \beta}{\|\beta\|_\Sigma} - \frac{1}{2}\frac{\Delta_{\text{core}}^\top \beta}{\|\beta\|_\Sigma},$$

$$-z = -\frac{1}{2}\frac{(\Delta_{\text{spur}}+\Delta_{\text{core}})^\top \beta}{\|\beta\|_\Sigma} = -\frac{1}{2}\frac{\Delta_{\text{spur}}^\top \beta}{\|\beta\|_\Sigma} - \frac{1}{2}\frac{\Delta_{\text{core}}^\top \beta}{\|\beta\|_\Sigma}.$$

Since $E_{\text{wge}} = \Phi(\max\{-z, w\})$, we obtain

$$E_{\text{wge}} = \Phi\left(\pm\frac{1}{2}\frac{\beta^\top \Delta_{\text{spur}}}{\|\beta\|_\Sigma} - \frac{1}{2}\frac{\beta^\top \Delta_{\text{core}}}{\|\beta\|_\Sigma}\right).$$

where $+$ corresponds to $w$ (unseen group is worst) and $-$ corresponds to $-z$ (seen group is worst). With Proposition 1 and the identity

$$\frac{\beta^{*\top} v}{\|\beta^*\|_\Sigma} = \text{cor}(\beta^*, v)\,\|v\|_\Sigma,$$

The claimed expression follows. $\qquad\qquad\qquad\qquad\qquad\qquad\qquad\qquad\qquad\qquad\qquad\square$

**Remark** The sign $\pm$ appearing in Theorem 1 reflects which subgroup constitutes the worst group. Specifically, since $E_{\text{wge}} = \Phi(\max\{-z, w\})$ where $z = \frac{\tilde{\Delta}^\top \beta}{2\|\beta\|_\Sigma} > 0$ and $w = \frac{(\frac{1}{2}\Delta_{\text{spur}}-\frac{1}{2}\Delta_{\text{core}})^\top \beta}{\|\beta\|_\Sigma}$:

- $+$ sign: $\Delta_{\text{spur}}^\top \beta^* > 0$, so $w > -z$ and the *unseen* subgroups $(y_{-1}, d_R), (y_{+1}, d_G)$ are the worst group.
- $-$ sign: $\Delta_{\text{spur}}^\top \beta^* < 0$, so $-z > w$ and the *seen* subgroups $(y_{+1}, d_R), (y_{-1}, d_G)$ are the worst group.

By Proposition 1, $\beta^* \propto \Sigma^{-1}\tilde{\Delta}$ implies $\Delta_{\text{spur}}^\top \beta^* \propto \|\Delta_{\text{spur}}\|^2_{\Sigma^{-1}} + \Delta_{\text{spur}}^\top \Sigma^{-1}\Delta_{\text{core}} > 0$ in the typical regime, so the $+$ sign (unseen groups worst) applies under ERM with spurious correlations.

**Remark** While the extreme correlation assumption is strong, such settings have been widely adopted in prior theoretical studies to isolate and analyze the effect of spurious correlations in a tractable manner (Yao et al., 2022; Lai & Muthukumar, 2024). Following this established tradition, we present our analysis and worst-group error decomposition under these extreme conditions. Empirically, we demonstrate in Section 4 that SCER remains effective even under less extreme correlation regimes, suggesting that the insights derived from this theoretical framework extend to more realistic settings.

FROM THEORETICAL ANALYSIS TO EMBEDDING LOSS

From Theorem 1, the worst-group error under the Gaussian mixture model can be expressed as

$$E_{\text{wge}} = \Phi\left(\pm\tfrac{1}{2}\operatorname{cor}(\beta^*, \Delta_{\text{spur}}) \|\Delta_{\text{spur}}\|_\Sigma - \tfrac{1}{2}\operatorname{cor}(\beta^*, \Delta_{\text{core}}) \|\Delta_{\text{core}}\|_\Sigma\right), \tag{4}$$

where $\Phi(\cdot)$ is the standard Gaussian CDF.

Equation 4 shows that two competing components govern the worst-group error:

$$\underbrace{\operatorname{cor}(\beta^*, \Delta_{\text{spur}}) \|\Delta_{\text{spur}}\|_\Sigma}_{\text{spurious term}} \quad \text{and} \quad \underbrace{\operatorname{cor}(\beta^*, \Delta_{\text{core}}) \|\Delta_{\text{core}}\|_\Sigma}_{\text{core term}}.$$

The first term increases the error by aligning with spurious directions, while the second decreases the error by aligning with core directions. Thus, minimizing $E_{\text{wge}}$ requires simultaneously suppressing the spurious term and enhancing the core term.

Motivated by this decomposition, we define the following surrogate regularizers:

$$\mathcal{L}_{\text{spur}} = \operatorname{cor}(\beta, \Delta_{\text{spur}}) \|\Delta_{\text{spur}}\|_\Sigma,$$
$$\mathcal{L}_{\text{core}} = \operatorname{cor}(\beta, \Delta_{\text{core}}) \|\Delta_{\text{core}}\|_\Sigma,$$

where $\beta$ denotes the classifier weights. The embedding-level regularization is then given by

$$\mathcal{L}_{\text{embedding}} = \lambda_{\text{spur}} \mathcal{L}_{\text{spur}} - \lambda_{\text{core}} \mathcal{L}_{\text{core}}. \tag{5}$$

**Remark.** In Theorem 1, we derived Equation 4 under the strict ERM setting with Gaussian assumptions, where the optimal $\beta^*$ has the closed form $\beta^* \propto \Sigma^{-1}\tilde{\Delta}$. In practice, however, $\beta$ corresponds to the weight parameters of a deep neural classifier trained via stochastic optimization. While $\beta$ may not coincide with the ERM minimizer, the decomposition in Equation 4 remains structurally valid: worst-group error is still determined by the trade-off between spurious and core alignment. This observation motivates applying the embedding regularization loss in Equation 5 to deep models.

## B  EXPERIMENTAL SETUP

### B.1  EXPERIMENTAL SETTING

This section provides comprehensive details of our experimental configuration for evaluating the SCER framework. We describe the datasets used for evaluation, the model architectures employed across different domains, the baseline methods used for comparison, the hyperparameter settings, and the evaluation metrics. These detailed specifications ensure reproducibility and provide a complete context for the results presented in the main paper.

#### B.1.1  DATASETS

We conduct experiments on six benchmark datasets. Four are image datasets, including Waterbirds, CelebA, MetaShift, and ColorMNIST, and two are text datasets, including CivilComments and MultiNLI. These datasets are established benchmarks for evaluating model robustness against spurious correlations. Dataset characteristics and statistics are summarized in Table 8.

**Waterbirds**  Waterbirds (Sagawa et al.) is a binary classification dataset constructed by superimposing bird images from the CUB dataset (Wah et al., 2011) onto backgrounds from the Places dataset (Zhou et al., 2017). The task involves classifying images as landbirds or waterbirds, where background type creates spurious correlations between land backgrounds and landbirds, and water backgrounds and waterbirds. We use the standard data splits from Idrissi et al. (2022).

**CelebA**  CelebA (Liu et al., 2015) contains approximately 200,000 celebrity images. We formulate the problem as a binary classification task, classifying hair as either blond or non-blond, with gender serving as the spurious attribute. We follow the standard splits from Idrissi et al. (2022).

Table 8: Statistics of datasets. For ColorMNIST, $\rho$ denotes the spurious correlation level between label $y$ and domain attribute (color) in the training set. A higher $\rho$ indicates stronger spurious correlation, making color a more predictive but misleading cue for classification. "One Subpopulation Omitted" represents an extreme case in which one group combination is completely absent from the training data. Groups are defined as combinations of class labels and attribute values, and Max/Min group columns show the sample sizes of the largest and smallest groups, respectively. The disparity between these values indicates the degree of group imbalance, which tests model robustness under uneven data distribution.

| Dataset | Data type | # Attr. | # Classes | # Train | # Val. | # Test | Max group | Min group |
|---|---|---|---|---|---|---|---|---|
| Waterbirds | Image | 2 | 2 | 4,795 | 1,199 | 5,794 | 3,498 | 56 |
| CelebA | Image | 2 | 2 | 162,770 | 19,867 | 19,962 | 71,629 | 1,387 |
| MetaShift | Image | 2 | 2 | 2,276 | 349 | 874 | 789 | 196 |
| **ColorMNIST** | | | | | | | | |
| $\rho = 80\%$ | Image | 2 | 2 | 30,000 | 10,000 | 20,000 | 12,228 | 3,002 |
| $\rho = 90\%$ | Image | 2 | 2 | 30,000 | 10,000 | 20,000 | 13,756 | 1,469 |
| $\rho = 95\%$ | Image | 2 | 2 | 30,000 | 10,000 | 20,000 | 14,518 | 731 |
| $\rho = 99\%$ | Image | 2 | 2 | 30,000 | 10,000 | 20,000 | 15,111 | 138 |
| One Subpopulation Omitted | Image | 2 | 2 | 30,000 | 10,000 | 20,000 | 15,249 | 0 |
| Two train envs | Image | 2 | 2 | 50,000 | 0 | 10,000 | 21,529 | 3,644 |
| - env1 ($\rho = 80\%$) | Image | 2 | 2 | 25,000 | 0 | 0 | 10,117 | 2,415 |
| - env2 ($\rho = 90\%$) | Image | 2 | 2 | 25,000 | 0 | 0 | 11,412 | 1,229 |
| CivilComments | Text | 8 | 2 | 148,304 | 24,278 | 71,854 | 31,282 | 1,003 |
| CivilComments (4 groups) | Text | 2 | 2 | 269,038 | 45,180 | 133,782 | 148,186 | 12,731 |
| MultiNLI | Text | 2 | 3 | 206,175 | 82,462 | 123,712 | 67,376 | 1,521 |

**MetaShift**    From the MetaShift benchmark (Liang & Zou, 2022), we use the Cats vs. Dogs binary classification task, where background type serves as the spurious attribute through correlations between indoor backgrounds with cats and outdoor backgrounds with dogs. We adopt the unmixed version provided by the dataset authors.

**ColorMNIST**    ColorMNIST (Arjovsky et al., 2019) is a synthetic variant of MNIST where each digit is assigned a color. The classification task groups digits into two categories: zero and four versus five and nine, with color serving as the spurious attribute through red and green assignments. Training data exhibits strong color-class correlations, validation data is balanced at 50:50, and test data reverses the correlation pattern to 10:90. Following Arjovsky et al. (2019), we inject 25% label noise and evaluate under varying spurious correlation strengths of 80:20, 90:10, and 95:5 ratios to assess robustness.

To test the limits of group imbalance, we conducted additional experiments on ColorMNIST with severe distributional skew, in which one subpopulation is completely omitted from training. Specifically, we use 15,249 samples for Label 0 with Color 0, 3,000 samples for Label 1 with Color 0, and 11,751 samples for Label 1 with Color 1, while Label 0 with Color 1 contains zero samples. This extreme setting evaluates model performance when specific subpopulations are entirely missing during training, representing a challenging scenario for robust classification methods.

To test the generalizability of SCER, we conducted additional experiments on ColorMNIST using the environment inference methodology from Creager et al. (2021). Following the prior protocol with two environments in train, we partition the dataset into 50,000 training samples and 10,000 test samples. The training set is further divided into two environments, each containing 25,000 samples that represent different spurious correlation patterns. We maintain consistency with our primary experimental setup by using identical model architectures, training procedures, and evaluation metrics across all experiments.

**CivilComments**    CivilComments (Borkan et al., 2019) provides a binary text classification benchmark designed to identify toxic language in online comments. Mentions of demographic groups serve as spurious attributes. We use the standard WILDS splits.

Table 9: Hyperparameters and search spaces.

| Model / Method | Hyperparameter | Default, Search Space |
|---|---|---|
| **Base Models** | | |
| ResNet | Learning rate | $0.001, 10^{\mathrm{Uniform}(-4, -2)}$ |
| | Batch size | $108, 2^{\mathrm{Uniform}(6, 7)}$ |
| BERT | Learning rate | $10^{-5}, 10^{\mathrm{Uniform}(-5.5, -4)}$ |
| | Batch size | $32, 2^{\mathrm{Uniform}(3, 5.5)}$ |
| | Dropout | $0.5$, `RandomChoice([0, 0.1, 0.5])` |
| **subpopulation Robustness Methods** | | |
| GroupDRO | $\eta$ | $0.01, 10^{\mathrm{Uniform}(-3, -1)}$ |
| CVaRDRO | $\alpha$ | $0.1, 10^{\mathrm{Uniform}(-2, 0)}$ |
| LfF | $q$ | $0.7$, $\mathrm{Uniform}(0.05, 0.95)$ |
| JTT | First-stage step fraction | $0.5$, $\mathrm{Uniform}(0.2, 0.8)$ |
| | $\lambda$ | $10, 10^{\mathrm{Uniform}(0, 2.5)}$ |
| LISA | $\alpha$ | $2, 10^{\mathrm{Uniform}(-1, 1)}$ |
| | $p_{\mathrm{select}}$ | $0.5$, $\mathrm{Uniform}(0, 1)$ |
| DFR | Regularization | $0.1, 10^{\mathrm{Uniform}(-2, 0.5)}$ |
| ElRep | $\theta_1, \theta_2$ | $0.1$, $\mathrm{Uniform}(0, 1)$ |
| **Domain-Invariant Methods** | | |
| IRM | $\lambda$ | $100, 10^{\mathrm{Uniform}(-1, 5)}$ |
| | Iterations of penalty annealing | $500, 10^{\mathrm{Uniform}(0, 4)}$ |
| MMD | $\gamma$ | $1, 10^{\mathrm{Uniform}(-1, 1)}$ |
| **Data Augmentation** | | |
| Mixup | $\alpha$ | $0.2, 10^{\mathrm{Uniform}(0, 4)}$ |
| **Class Imbalance Methods** | | |
| ReSample | – | Follow default implementation |
| ReWeight | – | Follow default implementation |
| SqrtWeight | – | Follow default implementation |
| Focal Loss | $\gamma$ | $1, 0.5 \times 10^{\mathrm{Uniform}(0, 1)}$ |
| CBLoss | $\beta$ | $0.9999, 1 - 10^{\mathrm{Uniform}(-5, -2)}$ |
| LDAM | max_m | $0.5, 10^{\mathrm{Uniform}(-1, -0.1)}$ |
| | scale | $30$, `RandomChoice([10, 30])` |
| **Proposed Method (SCER)** | | |
| SCER | $\eta$ | $0.01, 10^{\mathrm{Uniform}(-3, -1)}$ |
| | $\lambda_{\mathrm{core}}$ | $1$, $\mathrm{Uniform}(0, 1)$ |
| | $\lambda_{\mathrm{spur}}$ | $1$, $\mathrm{Uniform}(0, 1)$ |

**CivilComments (4 groups)** As an additional experiment on the CivilComments (Borkan et al., 2019) dataset, we use the coarse version for both training and evaluation to enable a fair comparison with GroupDRO-ES (Izmailov et al., 2022), which is known as the most competitive GroupDRO baseline and represents the state-of-the-art GroupDRO performance. In this setting, the spurious attribute is defined as 1 if the comment mentions at least one of the following categories, including male, female, LGBT, black, white, Christian, Muslim, or other religion. Otherwise, the spurious label is 0. The presence of these eight categories is spuriously correlated with the comment being classified as toxic. This configuration forms four groups, with two classes for toxic and non-toxic comments, and two attributes representing the presence or absence of the spurious attribute.

**MultiNLI** The MultiNLI dataset (Williams et al., 2018) addresses natural language inference by requiring models to classify the relationship between a premise and a hypothesis as entailment, contradiction, or neutrality. The presence of negation words serves as a confounding factor because they exhibit a strong correlation with contradiction labels. We adopt the data splits established in Idrissi et al. (2022).

Table 10: Summary of attribute availability settings across baseline methods.

| Train Attr. | Valid/Held-out Attr. | Method |
|---|---|---|
| Known | Known | GroupDRO, LISA, CVaRDRO, ReSample, PDE, ElRep(with GroupDRO) |
| Unknown | Known (Val) | JTT, LfF, CRT, SqrtReWeight |
| Unknown | Known (Held-out) | DFR |
| Unknown | Unknown | ReWeight, ERM, Mixup, MMD, Focal Loss, CB Loss, LDAM, Balanced Softmax |

### B.1.2 BASELINES

For fair comparison, we followed the training protocol of Gulrajani & Lopez-Paz, performing random search over each method's joint hyperparameter space. Hyperparameter configurations follow prior work (Yang et al., 2023b) for all baseline methods. We selected the hyperparameters that achieved the highest Worst Accuracy for each algorithm, then conducted three independent runs with different random seeds (0, 1, 2) to account for variance and ensure robustness of reported results. Final average performance and standard deviation are reported across these runs. Complete hyperparameter details are provided in Table 9, which summarizes the hyperparameters used for each baseline. In addition to standard optimization parameters such as batch size, learning rate, and dropout, we include key weighting parameters $\eta$, $\lambda_{spur}$, and $\lambda_{core}$ to suppress spurious features and enhance core features. These values are chosen to encourage the model to learn generalizable representations that generalize across distribution shifts.

We provide a clear summary in Table 10 indicating which baselines utilize group annotations at each stage: training, held-out selection, and validation. Following the evaluation protocol established in prior work (Yang et al., 2023b), we adopt the Known Attributes setting, in which group annotations are available for both the training and validation sets. All baselines are evaluated under this unified setting to ensure consistency and fair comparison. While some methods are designed to operate without group annotations during training or validation, all methods ultimately rely on group information for final model selection. Specifically, we select the best checkpoint for each method based on worst-group accuracy, ensuring alignment in model selection across all approaches. Thus, even when a method does not internally use group information during optimization, group annotations are consistently used for model selection to ensure fair and comparable evaluation.

To comprehensively evaluate our method's robustness, we compare its performance against a diverse set of baselines spanning standard ERM, subpopulation robustness methods, domain-invariant approaches, data augmentation techniques, and class imbalance methods. The primary baseline results appear in Tables 1, 2, 3, and 5 in the main text, with complete results for all baselines provided in Tables 11, 12, 13 and 15. Our baseline selection follows evaluation criteria established in recent studies (Deng et al., 2023; Wen et al., 2025), with particular attention to whether each method utilizes group annotations during training.

To ensure fair comparison, the main text reports only methods that rely on group annotations throughout the entire training and validation pipeline, while methods not satisfying this criterion are presented separately in the appendix to reflect differences in evaluation settings.

**Empirical Risk Minimization** ERM minimizes the average loss over all training samples without explicitly addressing group-specific performance disparities.

**subpopulation Robustness Methods** GroupDRO (Sagawa et al.) is a representative approach to distributional robustness that improves worst-group performance by assigning greater importance to groups with higher losses. CVaRDRO (Duchi & Namkoong, 2021) extends this approach to the individual sample level, enabling more fine-grained adjustment by assigning importance weights to individual samples. LfF (Nam et al., 2020) employs a co-training approach that simultaneously trains a biased model and a debiased model, guiding the model to learn by distinguishing between biased and unbiased features. JTT (Liu et al., 2021) is a two-stage strategy that identifies complex samples using information obtained from initial standard empirical risk minimization (ERM) training, then improves model robustness by increasing the weights of these samples in the second training stage. LISA (Yao et al., 2022) utilizes data augmentation techniques that mix data within and across attributes, encouraging the model to learn invariant features of attributes. DFR (Kirichenko et al.) is an efficient method that improves robustness by retraining only the last layer using a balanced validation set, rather than the entire model. PDE (Deng et al., 2023) employs a strategy of progressively

Table 11: Complete performance comparison on multiple datasets, each exhibiting different levels of spurious correlations. † denotes the performance reported from Yang et al. (2023b) except for ColorMNIST. ‡ denotes the performance reported from Deng et al. (2023) for Waterbird and CelebA. †† denotes the performance reported from Wen et al. (2025) for Waterbird and CelebA. For the remaining datasets, the detailed experimental settings follow Yang et al. (2023b), as described in Appendix B.1.2. SCER achieves the highest worst-group accuracy in Waterbirds, CelebA, MetaShift, and ColorMNIST, demonstrating its robustness in subpopulation shift scenarios.

| Algorithm | Waterbirds | | CelebA | | MetaShift | | ColorMNIST ($\rho = 80\%$) | |
|---|---|---|---|---|---|---|---|---|
| | Avg Acc | Worst Acc | Avg Acc | Worst Acc | Avg Acc | Worst Acc | Avg Acc | Worst Acc |
| ERM† | 84.1 ±1.7 | 69.1 ±4.7 | 95.1 ±0.2 | 62.6 ±1.5 | 91.3 ±0.3 | 82.6 ±0.4 | 38.2 ±2.4 | 30.9 ±2.7 |
| Mixup† | 89.5 ±0.4 | 78.2 ±0.4 | 95.4 ±0.1 | 57.8 ±0.8 | 91.6 ±0.3 | 81.0 ±0.8 | 41.2 ±14.0 | 12.4 ±2.0 |
| GroupDRO† | 88.8 ±1.8 | 78.6 ±1.0 | 91.4 ±0.6 | 89.0 ±0.7 | 91.0 ±0.1 | 85.6 ±0.4 | 73.5 ±0.3 | 73.1 ±0.1 |
| IRM† | 88.4 ±0.1 | 74.5 ±1.5 | 94.7 ±0.8 | 63.0 ±2.5 | 91.8 ±0.4 | 83.0 ±0.1 | 51.6 ±11.0 | 25.0 ±6.4 |
| CVaRDRO† | 89.8 ±0.4 | 75.5 ±2.2 | 95.2 ±0.1 | 64.1 ±2.8 | 92.1 ±0.2 | 84.6 ±0.0 | 42.7 ±5.9 | 30.9 ±3.5 |
| JTT† | 88.8 ±0.6 | 72.0 ±0.3 | 90.4 ±2.3 | 70.0 ±10.2 | 91.2 ±0.5 | 83.6 ±0.4 | 69.1 ±1.9 | 65.2 ±4.2 |
| LfF† | 87.0 ±0.3 | 75.2 ±0.7 | 81.1 ±5.6 | 53.0 ±4.3 | 80.2 ±0.3 | 73.1 ±1.6 | 74.0 ±4.1 | 4.1 ±2.3 |
| LISA† | 92.8 ±0.2 | 88.7 ±0.6 | 92.6 ±0.1 | 86.3 ±1.2 | 89.5 ±0.4 | 84.1 ±0.4 | 73.8 ±0.1 | 73.2 ±0.1 |
| MMD† | 93.0 ±0.1 | 83.9 ±1.4 | 92.5 ±0.7 | 24.4 ±2.0 | 89.4 ±0.1 | 85.9 ±0.7 | 41.7 ±0.7 | 27.5 ±7.0 |
| ReSample† | 89.4 ±0.9 | 77.7 ±1.2 | 92.0 ±0.8 | 87.4 ±0.8 | 91.2 ±0.1 | 85.6 ±0.4 | 73.7 ±0.1 | 72.2 ±0.3 |
| ReWeight† | 91.8 ±0.2 | 86.9 ±0.7 | 91.9 ±0.5 | 89.7 ±0.2 | 91.7 ±0.4 | 85.6 ±0.4 | 74.1 ±0.1 | 73.4 ±0.1 |
| SqrtReWeight† | 88.7 ±0.3 | 78.6 ±0.1 | 93.6 ±0.1 | 82.4 ±0.5 | 91.5 ±0.2 | 84.6 ±0.7 | 69.4 ±0.7 | 66.7 ±2.0 |
| CBLoss† | 91.3 ±0.7 | 86.2 ±0.3 | 91.2 ±0.7 | 89.4 ±0.7 | 91.7 ±0.4 | 85.5 ±0.4 | 73.6 ±0.2 | 72.8 ±0.2 |
| Focal† | 89.3 ±0.2 | 71.6 ±0.8 | 94.9 ±0.3 | 59.1 ±2.0 | 91.7 ±0.2 | 81.5 ±0.0 | 40.7 ±4.6 | 31.2 ±3.8 |
| LDAM† | 87.3 ±0.5 | 71.0 ±1.8 | 94.5 ±0.2 | 59.6 ±2.4 | 91.5 ±0.1 | 83.6 ±0.4 | 39.3 ±3.1 | 26.9 ±6.1 |
| BSoftmax† | 88.4 ±1.3 | 74.1 ±0.9 | 91.9 ±0.1 | 83.3 ±0.5 | 91.6 ±0.2 | 83.1 ±0.7 | 40.1 ±1.8 | 33.3 ±2.5 |
| DFR† | 92.3 ±0.2 | 91.0 ±0.3 | 91.9 ±0.1 | 90.4 ±0.1 | 88.4 ±0.3 | 85.4 ±0.4 | 68.4 ±0.1 | 67.3 ±0.2 |
| PDE‡ | 92.4 ±0.8 | 90.3 ±0.3 | 92.4 ±0.8 | 91.0 ±0.4 | 87.4 ±0.1 | 78.1 ±0.1 | 76.6 ±0.8 | 72.9 ±0.3 |
| ElRep†† | 92.9 ±0.7 | 88.8 ±0.7 | 92.8 ±0.2 | 91.4 ±1.0 | 85.9 ± 0.6 | 72.1 ± 2.5 | 50.3 ± 0.6 | 46.5 ± 4.3 |
| SCER | 92.1±0.2 | **91.2 ±0.2** | 92.7 ±0.2 | **91.4 ±0.1** | 91.6 ±0.3 | **86.7 ±0.8** | 74.1 ±0.1 | **73.6 ±0.2** |

expanding training data, guiding the model to first focus on core features and subsequently learn spurious features. Finally, ElRep (Wen et al., 2025) introduces both Nuclear-norm and Frobenius-norm regularization to representation vectors, enabling robust group generalization representation learning against spurious correlations by simultaneously emphasizing important features and maintaining diversity.

**Domain-Invariant Methods**    IRM (Arjovsky et al., 2019) is a method for finding invariant predictors that work commonly across multiple domains. Specifically, it trains feature extractors so that a fixed predictor becomes optimal in each domain, enabling the model to capture important characteristics effectively regardless of domain changes. This approach ensures robust performance even when domains change. MMD (Li et al., 2018) is a method that numerically measures the difference in feature distributions across domains and trains to reduce this difference, thereby mitigating distribution mismatches. In other words, it is an approach that aims to improve generalization performance by making feature representations across domains as similar as possible.

**Data Augmentation**    Mixup (Zhang et al., 2018) is a technique that generates synthetic training data by linearly interpolating two randomly sampled pairs of inputs and their labels. This smooths decision boundaries and serves as regularization to improve generalization performance and prevent overfitting. It is a simple yet effective data augmentation method that artificially expands the diversity of the training data by learning intermediate points between samples in the data space.

**Class Imbalance Methods**    ReSample (Japkowicz, 2000) mitigates class imbalance in training data through resampling techniques. ReWeight (Japkowicz, 2000) assigns weights to the loss function proportional to class frequency. SQRTWeight is another loss adjustment method that adjusts loss weights using the inverse square root of class frequencies, thereby increasing the influence of minority classes in imbalanced data. Focal Loss (Lin et al., 2017) focuses on difficult samples by reducing the contribution of easy-to-learn samples. CB Loss (Cui et al., 2019) calculates sample weights based on effective class size. LDAM (Cao et al., 2019) assigns distinct margins to each class to balance the decision boundaries. Balanced Softmax (Ren et al., 2020) modifies the softmax function itself to accommodate class imbalance.

Table 12: Complete performance comparison on ColorMNIST under increasing spurious correlation levels. SCER achieves the highest worst-group accuracy across different spurious correlation levels, demonstrating robustness to spurious feature reliance in ColorMNIST.

| | ColorMNIST | | | | | | | |
|---|---|---|---|---|---|---|---|---|
| | $\rho = 80\%$ | | $\rho = 90\%$ | | $\rho = 95\%$ | | $\rho = 99\%$ | |
| Algorithm | Avg Acc | Worst Acc | Avg Acc | Worst Acc | Avg Acc | Worst Acc | Avg Acc | Worst Acc |
| ERM | 38.2 ±2.4 | 30.9 ±2.7 | 31.9 ±7.4 | 7.8 ±3.4 | 41.1 ±13.0 | 10.1 ±4.5 | 50.7 ±8.2 | 8.5 ±5.0 |
| GroupDRO | 73.5 ±0.3 | 73.1 ±0.1 | 73.5 ±0.0 | 72.7 ±0.3 | 72.3 ±0.1 | 70.7 ±0.2 | 50.3 ±3.9 | 38.7 ±0.8 |
| LISA | 73.8 ±0.1 | 73.2 ±0.1 | 73.3 ±0.1 | 72.9 ±0.2 | 72.6 ±0.4 | 71.4 ±0.6 | 53.1 ±7.2 | 10.2 ±8.1 |
| ReSample | 73.7 ±0.1 | 72.2 ±0.3 | 72.6 ±0.5 | 70.8 ±1.6 | 71.8 ±0.1 | 70.7 ±0.5 | 57.1 ±3.0 | 45.4 ±10.0 |
| ReWeight | 74.1 ±0.1 | 73.4 ±0.1 | 73.6 ±0.0 | 72.8 ±0.2 | 72.4 ±0.5 | 70.7 ±1.1 | – | – |
| CBLoss | 73.6 ±0.2 | 72.8 ±0.2 | 72.7 ±0.2 | 71.7 ±0.1 | 71.8 ±0.3 | 70.8 ±0.3 | – | – |
| PDE | 76.6 ±0.8 | 72.9 ±0.3 | 72.6 ±0.2 | 70.2 ±0.7 | 73.4 ±0.4 | 70.0 ±0.3 | 56.3 ±4.3 | 47.5 ±8.9 |
| SCER | 74.1 ±0.1 | **73.6 ±0.2** | 73.6 ±0.1 | **73.0 ±0.1** | 73.5 ±0.3 | **72.8 ±0.3** | 61.0 ±2.0 | **56.0 ±2.2** |

## C  ADDITIONAL EXPERIMENT

This section presents supplementary experimental results that further validate the effectiveness and robustness of the SCER framework.

### C.1  QUANTITATIVE ANALYSIS

Here, we present the complete results for quantitative analyses omitted from the main text.

Table 13: Complete performance on ColorMNIST with one group absent. SCER achieves the highest worst-group accuracy.

| ColorMNIST (One Subpopulation Omitted) | | |
|---|---|---|
| Algorithm | Avg Acc | Worst Acc |
| ERM | 52.9 ±8.4 | 9.7 ±5.4 |
| GroupDRO | 53.6 ±2.5 | 44.1 ±6.2 |
| LISA | 49.5 ±2.4 | 13.5 ±11.0 |
| ReSample | 49.1 ±0.8 | 16.3 ±13.0 |
| ReWeight | 52.1 ±4.7 | 10.0 ±8.2 |
| CBLoss | 53.0 ±3.2 | 13.3 ±10.0 |
| PDE | 71.0 ±12.6 | 8.3±13.9 |
| SCER | 65.3 ±1.0 | **59.6 ±1.0** |

**Image Dataset**  Table 11 reports the complete results for all baselines on image datasets. Across this comprehensive set of methods, SCER consistently achieves the highest worst-group accuracy, demonstrating its superior robustness compared to all competing approaches.

**Image Dataset with Stronger Spurious Correlations**  Table 12 reports the complete results on ColorMNIST under varying spurious correlation levels, including all methods achieving worst-group accuracy above 70% in Table 11. SCER achieves the highest worst-group accuracy across most settings, confirming its robustness under subpopulation shift.

In addition to the standard evaluation settings in Table 12, we examine a more challenging scenario in which a minority group is completely absent during training. Table 13 reports the complete results under this extreme setting. As shown, SCER achieves the highest worst-group accuracy, significantly outperforming all competing methods. Notably, class imbalance methods such as ReWeight (Japkowicz, 2000) and CB Loss (Cui et al., 2019) exhibit substantial performance degradation when a group is entirely missing, similar to other approaches. In contrast, SCER maintains robust performance even under this extreme distributional shift, demonstrating its effectiveness in handling severe group imbalance.

Table 14: Performance comparison without explicit bias labels in noisy environments.

| ColorMNIST (Two train envs) | |
|---|---|
| Method | Avg Acc |
| IRM | 54.8 ±8.3 |
| EIIL (IRM) | 53.7 ±2.1 |
| EIIL + DRO | 54.6 ±5.9 |
| EIIL + SCER | **65.1 ±2.8** |

**Integration with Environment Inference Methods.**  Following Table 4, we additionally experimented to verify whether SCER maintains robustness when the environment partitioning accuracy is imperfect, at approximately 70%. All other settings remain identical to those in Table 4, and the results are presented in Table 14. Even under reduced environment partitioning accuracy, EIIL + SCER achieves an average accuracy of 65.1%, outperforming both EIIL + DRO at 54.6% and EIIL with IRM at 53.7%. This suggests that SCER operates effectively even under imperfect environment partitioning conditions.

**Text Dataset**  Table 15 presents the complete performance comparison on text datasets, including CivilComments and MultiNLI. SCER consistently achieves superior worst-group accuracy across these benchmarks, demonstrating its effectiveness beyond image domains. On CivilComments, SCER attains a worst-group accuracy of 74.0% . Similarly, on MultiNLI, SCER achieves the best worst-group accuracy of 76.8%.

Tables 11, 12, 13, and 15 demonstrate that SCER consistently achieves superior performance across diverse benchmark datasets and in comparison with various state-of-the-art algorithms. Notably, SCER achieves robust performance not only on vision tasks but also on natural language understanding tasks, confirming that our mechanism of separating spurious and core features at the embedding level generalizes effectively across domains.

Table 15: Complete performance comparison on text datasets. [‡] from Deng et al. (2023) for Civilcomments. [††] from Wen et al. (2025) for Civilcomments. SCER achieves the best worst-group accuracy across multi-class and domain settings.

| Algorithm | CivilComments | | MultiNLI | |
|---|---|---|---|---|
| | Avg Acc | Worst Acc | Avg Acc | Worst Acc |
| ERM[†] | 85.4 ±0.2 | 63.7 ±1.1 | 80.9 ±0.1 | 66.8 ±0.5 |
| Mixup[†] | 84.9 ±0.3 | 66.1 ±1.3 | 81.4 ±0.3 | 68.5 ±0.6 |
| GroupDRO[†] | 81.8 ±0.6 | 70.6 ±1.2 | 81.1 ±0.3 | 76.0 ±0.7 |
| IRM[†] | 85.5 ±0.0 | 63.2 ±0.8 | 77.8 ±0.6 | 63.6 ±1.3 |
| CVaRDRO[†] | 83.5 ±0.3 | 68.7 ±1.3 | 75.1 ±0.1 | 63.0 ±1.5 |
| JTT[†] | 83.3 ±0.1 | 64.3 ±1.5 | 80.9 ±0.5 | 69.1 ±0.1 |
| LfF[†] | 65.5 ±5.6 | 51.0 ±6.1 | 71.7 ±1.1 | 63.6 ±2.9 |
| LISA[†] | 82.7 ±0.1 | 73.7 ±0.3 | 80.3 ±0.4 | 73.3 ±1.0 |
| MMD[†] | 84.6 ±0.2 | 54.5 ±1.4 | 78.8 ±0.1 | 69.1 ±1.5 |
| ReSample[†] | 82.2 ±0.0 | 73.3 ±0.5 | 77.2 ±0.2 | 72.3 ±0.8 |
| ReWeight[†] | 82.5 ±0.0 | 72.5 ±0.0 | 81.0 ±0.2 | 68.8 ±0.4 |
| SqrtReWeight[†] | 83.3 ±0.5 | 71.7 ±0.4 | 80.7 ±0.3 | 69.5 ±0.7 |
| CBLoss[†] | 82.9 ±0.1 | 73.3 ±0.2 | 80.6 ±0.1 | 72.2 ±0.3 |
| Focal[†] | 85.5 ±0.2 | 62.0 ±1.0 | 80.7 ±0.2 | 69.4 ±0.7 |
| LDAM[†] | 81.9 ±2.2 | 37.4 ±8.1 | 80.7 ±0.3 | 69.6 ±1.6 |
| BSoftmax[†] | 83.8 ±0.0 | 71.2 ±0.4 | 80.9 ±0.1 | 66.9 ±0.4 |
| DFR[†] | 83.3 ±0.0 | 69.6 ±0.2 | 81.7 ±0.0 | 68.5 ±0.2 |
| PDE[‡] | 86.3 ±1.7 | 71.5 ±0.5 | 69.1 ±0.3 | 65.8 ±0.6 |
| ElRep[††] | 79.0 ±0.7 | 70.5 ±0.5 | 69.0 ±3.8 | 66.8 ±5.2 |
| SCER | 81.7 ±0.4 | **74.0 ±1.0** | 80.4 ±0.1 | **76.8 ±0.5** |

Table 16: Performance comparison on Waterbirds, CelebA, MultiNLI, and CivilComments (4 groups) with GroupDro-ES. [§] from Izmailov et al. (2022). Experimental and dataset settings also follow Izmailov et al. (2022). SCER achieves the best worst-group accuracy across all datasets.

| Algorithm | Waterbirds Worst Acc | CelebA Worst Acc | MultiNLI Worst Acc | CivilComments (4 groups) Worst Acc |
|---|---|---|---|---|
| GroupDRO-ES[§] | $90.7 \pm 0.6$ | $90.6 \pm 1.6$ | 73.5 | 80.4 |
| SCER | $\mathbf{91.7 \pm 0.1}$ | $\mathbf{91.9 \pm 0.3}$ | **77.9** | **82.9** |

**Deep Analysis with GroupDRO**  SCER proposes a novel objective function that decomposes worst-group error into spurious and core components, then directly regularizes the embedding space to suppress spurious correlations while enhancing core features. To validate the effectiveness of this approach, we conducted an in-depth comparison with GroupDRO. Since early stopping is well known to be essential for achieving optimal GroupDRO performance (Izmailov et al., 2022), we compared SCER with GroupDRO-ES (Izmailov et al., 2022), which serves as the most competitive GroupDRO baseline. Note that for a fair and direct comparison with GroupDRO-ES, we adopted the experimental setup from Izmailov et al. (2022) and used their reported results for GroupDRO-ES, which differ from the configuration in our main experiments following Yang et al. (2023b).

As shown in Table 16, SCER achieves consistent performance improvements over GroupDRO-ES across all datasets. Specifically, SCER demonstrates gains of 1.0% on Waterbirds and 1.3% on CelebA, with a particularly notable 4.4% improvement on MultiNLI. This represents meaningful progress, given that most group-robust methods struggle to exceed worst-group accuracy in the low 70% range on this dataset. For CivilComments, while Tables 5 and 15 follow the 16 group setting with two classes and eight attributes as in prior work (Yang et al., 2023b), we also conducted experiments using the 4 group setting with two classes and two attributes in Table 16 to allow a fair comparison with (Izmailov et al., 2022), achieving a 2.5% improvement. Detailed information about the dataset configurations is provided in Table 8.

Beyond quantitative gains, we provide qualitative evidence that SCER fundamentally restructures the embedding space. Figure 3 shows through t-SNE analysis that while ERM and GroupDRO separate clusters by background, SCER achieves label-based clustering independent of spurious features, confirming domain-invariant representations. Figure 5 further demonstrates via Grad-CAM

that SCER attends to target objects rather than backgrounds, showing that our embedding regularization yields genuine improvements in representation learning.

We also conduct a comprehensive quantitative analysis to evaluate the computational complexity and scalability of the SCER framework. These experiments provide detailed insights into SCER's adaptation mechanisms across various scenarios and assess its practical feasibility for real-world deployment.

Table 17: Average per-iteration training time (seconds).

| Method | Avg One Iteration (s) |
|---|---|
| ERM | 0.0654 |
| GroupDRO | 0.0915 |
| DFR | 0.1025 |
| SCER | 0.0988 |

**Complexity analysis** While our method requires computational investment for correlation metric and mean embedding computations, it maintains computational costs comparable to established robust training methods. As shown in Table 17, our approach requires 0.0988 seconds per iteration, which is marginally higher than GroupDRO at 0.0915 seconds but faster than the two-stage DFR method at 0.1025 seconds.

Importantly, DFR's computational overhead stems from its inherent two-stage paradigm (Kirichenko et al.): first training with ERM to learn representations, then retraining only the final layer with group-balanced data. This approach suffers from critical limitations, including the requirement for high-quality, balanced validation data and complete failure when ERM cannot capture core features, such as when minority groups are entirely absent from training data. In contrast, our single-stage approach achieves substantially better worst-group accuracy than these alternatives while avoiding DFR's structural constraints. Most importantly, our method requires no balanced validation data or multistage training, delivering superior robustness with greater practical simplicity.

Table 18: Sensitivity analysis of embedding regularization on CelebA. Worst-group accuracy remains relatively stable across settings, with the best results observed when moderate $\lambda_{core}$ is applied.

| **CelebA** | | | | | | | | |
|---|---|---|---|---|---|---|---|---|
| both $\lambda = 0$ | | | $\lambda_{core}$ (fixed $\lambda_{spur}$) | | | $\lambda_{spur}$ (fixed $\lambda_{core}$) | | |
| Setting | Avg Acc | Worst Acc | $\lambda_{core}$ | Avg Acc | Worst Acc | $\lambda_{spur}$ | Avg Acc | Worst Acc |
| 0 | 91.4 ±0.6 | 89.0 ±0.7 | 0.0 | 92.5 ±0.1 | 91.1 ±0.2 | 0.0 | 92.0 ±0.3 | **91.0 ±0.2** |
| – | – | – | 0.5 | 92.7 ±0.2 | **91.4 ±0.1** | 0.5 | 91.7 ±0.6 | 90.9 ±0.7 |
| – | – | – | 1.0 | 92.7 ±0.2 | 90.9 ±0.5 | 1.0 | 92.0 ±0.1 | 90.7 ±0.1 |

## C.2 QUALITATIVE ANALYSIS

We provide comprehensive quantitative evaluations, including additional sensitivity analysis across hyperparameters.

**Embedding Regularization Analysis** In addition to the results in Table 6, we conduct sensitivity analysis for $\lambda_{core}$ and $\lambda_{spur}$ on the large-scale real-world dataset CelebA. The results confirm the generalizability of our ColorMNIST findings beyond synthetic settings. CelebA exhibits even more pronounced performance improvements with appropriate hyperparameter values, highlighting the effectiveness of embedding regularization in mitigating spurious correlations in realistic scenarios, as shown in Table 18

## D STATEMENT ON THE USE OF LARGE LANGUAGE MODELS

Large Language Models (LLMs) were used in a limited capacity during the preparation of this research. Specifically, LLMs were used to check grammar and refine sentence structure after the initial draft was completed, primarily to correct awkward expressions and maintain consistency in writing style. However, all core research ideas, analytical methodologies, interpretations of the results, and conclusions were developed entirely by the authors. The LLM did not contribute to any creative content or academic judgments. This use of LLMs was conducted within limits that do not compromise the originality or academic integrity of the research.

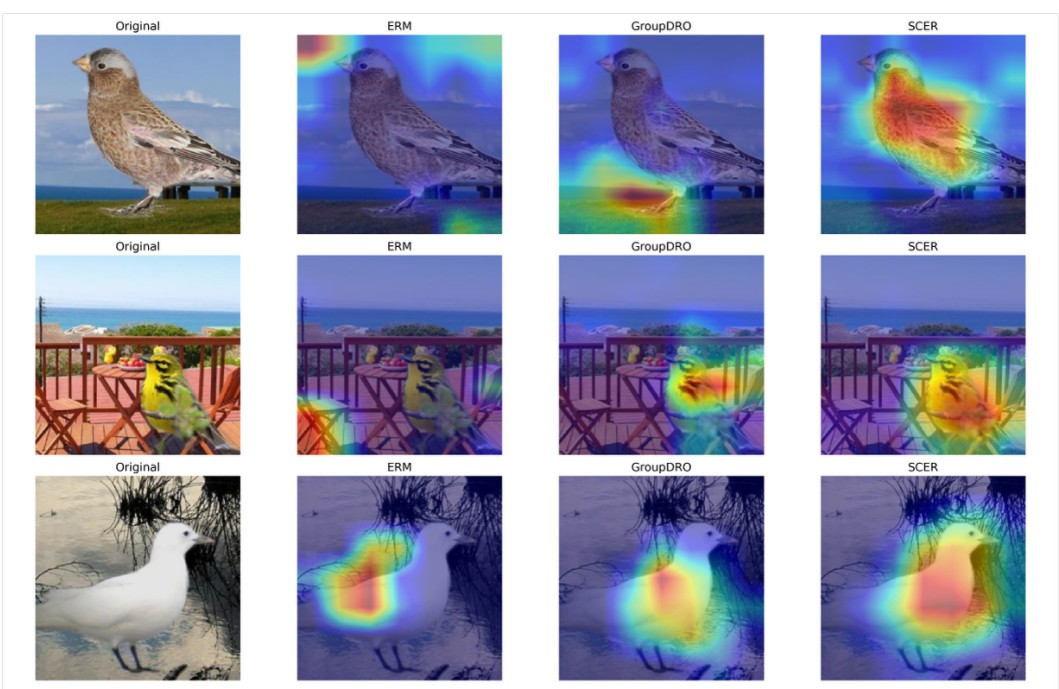

Figure 6: Comparison of Grad-CAM visualizations on the Waterbirds dataset. SCER directs attention to meaningful features, reducing focus on spurious regions.

