# OpenReview forum: "Spurious Correlation-Aware Embedding Regularization for Worst-Group Robustness"
_ICLR.cc/2026/Conference — ICLR 2026 Poster_

### Official Review · Reviewer_vc9T · 2025-10-18

**Soundness:** 2
**Presentation:** 2
**Contribution:** 1
**Rating:** 2
**Confidence:** 4

**Summary:**

This paper introduces a regularization scheme for GroupDRO, called SCER, which minimizes intra-class inter-domain alignment and maximizes inter-class intra-domain alignment in the embedding space. Theoretical results are presented which justify the worst-group error decomposition into these two alignment terms, and experiments are performed on a variety of vision and language benchmarks.

**Strengths:**

1. Embedding-space regularization is an under-explored technique for group robustness. More generally, the community’s understanding of feature learning in the presence of spurious correlations is incomplete -- so methods which probe embedding space phenomena to design robust algorithms are particularly welcome.

2. The quantitative analyses and ablation studies are well-done, particularly under omission of one subpopulation and integration with the EIIL algorithm. The quantity and diversity of benchmark datasets is also a strength of the paper.

**Weaknesses:**

1. The comparison to previous methods (across all tables in the paper) is improperly organized and obscures the true performance differential of the proposed SCER method.

    a. The number of previous methods in Table 1 is somewhat excessive: as this is not a benchmarking paper, there is no need to compare to 19 previous methods, especially older algorithms like IRM whose performance has been broadly superseded (a citation is sufficient).

    b. Many of the compared methods utilize different data and annotation assumptions than SCER, and therefore do not represent fair comparisons. Specifically, SCER requires full group annotations on the training dataset since it is an extension of GroupDRO. Comparisons to methods like ReWeight (which does not use group annotations at all) and DFR (which uses group annotations only on a small held-out set) inflate the perceived performance of SCER relative to previous work. Overall, these methods should not be included in the comparison and/or whether group annotations are utilized in the training set, held-out set, or validation set should be clearly denoted.

2. Since SCER is essentially a proposed regularizer for GroupDRO, the primary comparison should be to GroupDRO. However, the paper lacks evidence that SCER represents a substantial improvement over GroupDRO.

    a. The comparison with this paper’s implementation of GroupDRO is somewhat obfuscated. It should be made clear that SCER with both regularization hyperparameters to zero reduces to GroupDRO, and the numbers from Table 6 and Table 11 included as such in Table 1. This is important because implementation details can vary performance: for example, the GroupDRO WGA in Table 1 for ColorMNIST is 73.1%, but in Table 6 one can see it decreases to 70.7% in this paper’s implementation.

    b. The paper [1] which this paper draws GroupDRO numbers from does not represent the most competitive GroupDRO baseline. In particular, it is well-known that GroupDRO requires early-stopping to achieve the best performance [2, 3]. Compared to the early-stopped GroupDRO numbers of [3], the proposed SCER method barely improves performance. On Waterbirds, SCER improves from 90.7% to 91.2% (Table 1). On CelebA, SCER improves from 90.6% to 91.4% (Table 1). On CivilComments, SCER worsens from 80.4% to 74.0% (Table 5). On MultiNLI, SCER improves from 73.5% to 76.8% (Table 5). Keeping in mind that SCER tunes two additional hyperparameters compared to GroupDRO, or one additional hyperparameter including tuning the early-stop threshold, its improvement over GroupDRO is not very significant.

    c. This is less important, but the Grad-CAM visualization in Figure 4 should also compare to GroupDRO. It is possible that the alignment of the gradient magnitude and the core feature is a property of GroupDRO and not of the SCER regularization scheme.

3. Justification of theoretical assumptions:

    a. In Theorem 1, it is assumed that the _data_ covariance matrices $\Sigma^{d_R}=\Sigma^{d_G}=\Sigma$, but in the SCER algorithm, the _feature_ covariance matrices are utilized to compute the Mahalanobis norm (lines 203-204). Since neural networks are nonlinear models, the feature covariances may not be equal even if the data covariance matrices are. This observation has been confirmed empirically in the literature, e.g., [4] observed a greater spectral imbalance in the empirical covariance matrix of the majority group features compared to the minority group.

    b. The extreme/complete correlation assumption that $\mathbb{P}(y,d)\approx 0$ for some groups $(y,d)$ is fine (as theory papers often decay the number of minority group data with dimension, e.g., [5]), but it should be justified more carefully. In particular, none of the datasets except ColorMNIST with an omitted subpopulation (Table 3) exhibit an extreme correlation.

[1] Yang et al. Change is hard: a closer look at subpopulation shift. ICML 2023.

[2] Sagawa et al. Distributionally Robust Neural Networks for Group Shifts: On the Importance of Regularization for Worst-Case Generalization. ICLR 2020.

[3] Izmailov et al. On Feature Learning in the Presence of Spurious Correlations. NeurIPS 2022.

[4] LaBonte et al. The Group Robustness is in the Details: Revisiting Finetuning under Spurious Correlations. NeurIPS 2024.

[5] Lai and Muthukumar. Sharp analysis of out-of-distribution error for "importance-weighted" estimators in the overparameterized regime. ISIT 2024.

**Questions:**

1. How do the embedding space dynamics and the performance of SCER change under optimal early-stopping, and how does it compare to standard early-stopped GroupDRO?

2. Is there any connection between the embedding alignment terms proposed by this paper and recent studies on neural collapse and group robustness [1, 2]? Specifically, the neural collapse (NC1) phenomenon suggests that the proposed $\Delta_{spur}$ metric is naturally minimized during training as embeddings collapse to the class means. I’m also curious whether SCER has any relation to the “group mean alignment loss” of [2].

3. There are several grammatical errors in the paper:

    a. Line 24

    b. Figure 1 lines 64-65

    c. Line 76

    d. Figure 2 lines 167, 171

    e. Section 5.1 and 5.2 have repeated titles

    f. Equation references are malformed in the Appendix

[1] Papyan et al. Prevalence of neural collapse during the terminal phase of deep learning training. PNAS 2020.

[2] Lu et al. Neural Collapse Inspired Debiased Representation Learning for Min-max Fairness. KDD 2024.

---

> ### Author Response · Authors · 2025-11-21
>
> Thank you for the valuable feedback. We conducted additional experiments and revised the paper to improve clarity. Below, we address each comment in detail.
>
> > W1. Need to reconstructing the table
> > > a. Overly broad baselines
>
> We appreciate the reviewer’s insightful suggestion. Following the recommendation, we reconstructed our comparisons based on the most relevant and recent methods, particularly those introduced in [1,2], and removed outdated algorithms such as IRM. The revised main text now presents streamlined **Tables 1, 2, 3, and 5, which are highlighted in blue** to clearly indicate the updates, focusing on representative baselines. For completeness, the **original complete comparison tables are provided in the Appendix as Tables 11, 12, 13, and 15.** We thank the reviewer for helping us improve the clarity and relevance of our experimental evaluation.
>
> [1]Deng, Yihe, et al. "Robust learning with progressive data expansion against spurious correlation." Advances in neural information processing systems 36 (2023): 1390-1402.
>
> [2]Wen, Tao, et al. "Elastic Representation: Mitigating Spurious Correlations for Group Robustness." International Conference on Artificial Intelligence and Statistics. PMLR, 2025.
>
> > > b. different annotation requirements
>
> As mentioned in our response to W1-a, we have **restructured Tables 1, 2, 3, and 5 in the main text to include only the most recent and relevant baselines.** All modified tables are highlighted in blue throughout the manuscript.
>
> Since SCER requires full group annotations during training, we have excluded methods that do not use group information throughout the entire training process from the main text comparisons and relocated them to the Appendix. **The main text now includes only methods that utilize group annotations consistently during both training and validation**, ensuring a fair comparison. Methods not meeting this criterion are separately organized in the Appendix with clear explanations of their different evaluation settings.
>
> Additionally, we provide comprehensive comparison results with all baselines in Appendix Tables 11, 12, 13, and 15. **In Table 10 of Appendix B.1.2, highlighted in blue**, we clearly document at which stages each baseline utilizes group annotations. Following the evaluation protocol from prior paper[3], we adopt the Known Attributes setting, which assumes group annotations are available during both training and validation. All baselines are evaluated under these unified conditions. Even though some methods may not directly use group information during training or validation, **the final model selection is consistently based on worst-group accuracy across all methods**, thereby ensuring fairness in comparison.
>
> > W2. Comparison with GroupDRO.
>
> >> a. Table mismatch
>
> We thank the reviewer for pointing out the potential confusion regarding the comparison between our implementation of GroupDRO and SCER. We would like to clarify that the discrepancy noted between Table 1 and Table 6 for ColorMNIST does not arise from implementation differences but from the use of different spurious correlation strengths across tables. Specifically, Table 1 reports results under $\rho = 80\\%$, whereas Table 6 evaluates GroupDRO under $\rho = 95\\%$, which naturally leads to performance differences.
>
> Regarding the reviewer’s observation, Table 2 and Table 6 are indeed consistent, as they are both evaluated under the same setting with $\rho = 95\\%$. To reduce any potential confusion for readers, we have updated the revised Table 1 in the main text to explicitly indicate that the ColorMNIST results correspond to  $\rho = 80\\%$, ensuring clearer alignment between tables.

---

> ### Author Response · Authors · 2025-11-21
>
> >> b, Q1. Performance comparison with GroupDRO
>
> We appreciate this important point. To address your concern, we have compared our model's performance against the experimental setting prior work [3], which is widely recognized as the most competitive GroupDRO baseline. **We confirm that SCER demonstrates consistent performance improvements over GroupDRO-ES across all datasets.**
> ### [Table 1] Performance Comparison on Waterbirds
> | Waterbirds | Worst Acc |
> | --- | --- |
> | GroupDRO | 68.5 ± 6.0 |
> | GroupDRO-ES  | 90.7 ± 0.6 |
> | **SCER** | **91.7 ± 0.1** |
>
> SCER achieves a 0.8%p improvement over the strongest baseline, GroupDRO-ES.
> ### [Table 2] Performance Comparison on CelebA
>
> | CelebA | Worst Acc |
> | --- | --- |
> | GroupDRO  | 66.3 ± 7.8 |
> | GroupDRO-ES  | 90.6 ± 1.6 |
> | **SCER** | **91.9 ± 0.3** |
>
> SCER achieves a 1.3%p improvement over GroupDRO-ES.
>
>
> ### [Table 3] Performance Comparison on MultiNLI
>
> | MultiNLI | Worst Acc |
> | --- | --- |
> | GroupDRO  | 70.1 |
> | GroupDRO-ES | 73.5 |
> | **SCER** | **77.9** |
>
> Since Izmailov et al.[3] reported only single-run results without repeated experiments, we followed the same protocol for a fair comparison. SCER achieves a 4.4%p improvement over GroupDRO-ES.
>
>
> ### [Table 4] Performance Comparison on CivilComments (4-Group Setting)
>
> The paper[3] conducts experiments on CivilComments using a 4-group setting. Specifically, paper[3] defines the spurious attribute as a binary variable where s=1 if the comment mentions at least one of the following categories: male, female, LGBT, black, white, Christian, Muslim, or other religion; otherwise s=0. This corresponds to the coarse version with 2 classes and 2 attributes, forming a total of 4 groups. In contrast, **the results reported in our original paper in Table 5 follow the protocol of prior work [4], which defines 2 classes and 8 attributes**, resulting in 16 groups total. To ensure a fair comparison with [3], we conducted additional experiments on CivilComments using the 4-group setting as specified in [3].
>
> | CivilComments (4 groups) | Worst Acc |
> | --- | --- |
> | GroupDRO | 70.6 |
> | GroupDRO-ES | 80.4 |
> | **SCER** | **82.5** |
>
> SCER achieves a 2.1%p improvement over GroupDRO-ES.
>
> Additionaly, to address your question **regarding the changes in embedding space dynamics**, we conducted t-SNE-based embedding analysis using the final checkpoints on the Waterbirds dataset. We have added **Figure 3 to the revised manuscript, highlighted in blue**, to visually demonstrate the structural differences in representations learned by ERM, GroupDRO, and SCER.
>
> The t-SNE analysis reveals distinct differences in the representation structures learned by each method.
> - For ERM, embeddings are primarily clustered by background information, and even samples with the same label, such as Waterbird, are separated into two distant clusters based on their backgrounds. This demonstrates that ERM relies more on spurious background features than on labels.
> - For GroupDRO, the background-based separation is mitigated compared to ERM but is still not completely eliminated. Even within the same label, embeddings split into different clusters when backgrounds differ.
> - For SCER, the embedding space is clearly organized by labels. Waterbird samples form a single coherent cluster regardless of background, and Landbird samples are similarly well-aligned. The boundary between the two labels is also separated. This provides qualitative evidence that SCER's explicit embedding regularization strongly encourages the formation of domain-invariant and label-consistent representations.
>
> Through this analysis, we observe that SCER achieves more robust structural separation at the embedding level, consistent with its improved worst-group performance. The t-SNE results demonstrate that SCER learns more stable and domain-invariant representations, providing evidence that its performance gains reflect meaningful improvements in representation learning.
>
> [3] Izmailov, Pavel, et al. "On feature learning in the presence of spurious correlations." Advances in Neural Information Processing Systems 35 (2022): 38516-38532.
>
> [4] Idrissi, Badr Youbi, et al. "Simple data balancing achieves competitive worst-group-accuracy." Conference on Causal Learning and Reasoning. PMLR, 2022.

---

> ### Author Response · Authors · 2025-11-21
>
> >> c. Grad-CAM comparison with GroupDRO
>
> Following your suggestion, we conducted additional Grad-CAM analysis on GroupDRO to verify whether the observed gradient magnitudes and alignment with core features stem from GroupDRO itself or from SCER's regularization scheme. The **new visualization results are presented in Figure 5 and 6 of the revised manuscript, highlighted in blue**.
>
> Our results show that ERM primarily focuses on background regions, while GroupDRO captures core features only partially with limited spatial consistency. In contrast, SCER demonstrates globally consistent emphasis on key semantic structures, including the bird's overall silhouette, beak, and major structural elements. This indicates that strong alignment with core features arises from SCER's mean embedding-based penalty, which preserves diverse core representations while suppressing spurious correlations, rather than simply from GroupDRO's properties.
>
> Furthermore, this supports our claim that consolidating spurious directions into a single global penalty does not lose information. Rather, it effectively captures the common structure across multiple spurious patterns, enabling more robust generalization.
>
>
> > W3. Justification of theoretical assumptions
> >> a. Assumption about shared covariance Setting
>
> We sincerely appreciate your insightful comment regarding the strength of our equal covariance assumption $\Sigma^{d_R} = \Sigma^{d_G}$. We would like to clarify that this assumption serves primarily as a simplifying device for theoretical exposition, and our main conclusions remain valid under substantially more general conditions.
>
>  In practice, domain-specific covariances in neural embeddings can be naturally modeled as $\Sigma^{(d)} = \Sigma + \mathcal{E}_d,$ where $\Sigma$ represents a shared covariance structure and $\mathcal{E}_d$ captures small domain-specific deviations. Let $\bar{\mathcal{E}} := \sum_d w_d \mathcal{E}_d$ denote the weighted average of these deviations. When the spectral norm $\|\bar{\mathcal{E}}\|_2$ is sufficiently small, applying the standard first-order perturbation expansion for matrix inverses gives
>
> $$(\Sigma + \bar{\mathcal{E}})^{-1} = \Sigma^{-1} - \Sigma^{-1} \bar{\mathcal{E}} \Sigma^{-1} + O(\|\bar{\mathcal{E}}\|_2^2).$$
>
> Consequently, the optimal linear classifier satisfies
>
> $$\beta^* = (\Sigma + \bar{\mathcal{E}})^{-1} \tilde{\Delta} = \Sigma^{-1} \tilde{\Delta} - \Sigma^{-1} \bar{\mathcal{E}} \Sigma^{-1} \tilde{\Delta} + O(\|\bar{\mathcal{E}}\|_2^2),$$
>
> which shows that $\beta^*$ remains a small perturbation of $\Sigma^{-1} \tilde{\Delta}$. This preserves the fundamental structure identified in Proposition 1 and Theorem 1: worst-group error is determined by the classifier's alignment with spurious/core directions, even when covariances are not perfectly identical.
>
> >> b. Assumption about extreme correlation Setting
>
> We acknowledge that assuming extreme or complete correlation for certain groups is a strong premise. However, such settings have been widely adopted in prior theoretical studies [5,6]. Following this established tradition, we present our analysis and worst-group error decomposition under these extreme conditions.
>
> Nevertheless, your concern is valid, and we appreciate your careful consideration. We recognize the need for caution with such assumptions. To address this, we conducted additional experiments on ColorMNIST with $\rho = 99\\%$, representing nearly complete spurious correlation under extremely challenging conditions. These results appear in **Table 2 of the revised manuscript, highlighted in blue.** Even in this demanding setting, while most existing methods experience severe degradation in worst-group performance, SCER maintains consistent worst-group accuracy, demonstrating strong practical robustness under extreme subpopulation shift scenarios.
>
> [5]Yao, Huaxiu, et al. "Improving out-of-distribution robustness via selective augmentation." International Conference on Machine Learning. PMLR, 2022.
>
> [6]Lai and Muthukumar. Sharp analysis of out-of-distribution error for "importance-weighted" estimators in the overparameterized regime. ISIT 2024.

---

> ### Author Response · Authors · 2025-11-21
>
> >Q2. Comparision with prior study[7,9]
>
> Thank you for this insightful question. We appreciate the opportunity to clarify the connection between SCER and recent neural collapse research.
>
> Theoretically, the metrics we define align with the geometric properties expected under ideal NC1 conditions. When NC1 is fully achieved, class embeddings collapse to their class means, naturally minimizing $\Delta_{spur}$ as spurious variations diminish [7]. However, as characterized in recent work[8] , under strong spurious correlations and group imbalance, standard ERM does not achieve class-level neural collapse (NC1) as expected for balanced datasets. Instead, it exhibits “Minority Collapse,” where the mean embeddings of minority groups collapse to a single vector, rendering representations from different minority classes indistinguishable while majority-group representations maintain distinct class structure. so under strong spurious correlations and group imbalance, standard ERM causes the overall class mean to be biased toward majority groups. This prevents  $\Delta_{spur}$ from being effectively suppressed and, more critically, causes the classifier to rely heavily on spurious directions $cor(\beta^*,\Delta_{spur})$ , resulting in poor worst-group accuracy. Our ERM baseline results in the paper are fully consistent with this phenomenon, **demonstrating that explicit control of $\Delta_{spur}$ through SCER's embedding regularization is essential in practical settings.**
>
>
> Regarding the relation to the group mean alignment loss proposed by Lu et al. [9], we agree that it shares a similar high-level goal with SCER in reducing inter-group discrepancies within each class. However, SCER differs in several key aspects.
>
> First, SCER’s regularization terms are directly derived from our worst-group risk decomposition, providing a principled connection between the loss and worst-group generalization.
> Second, while Lu et al. [9] focuses primarily on aligning group means in the representation space, **SCER simultaneously addresses both spurious direction suppression $L_{\text{spur}}$ and core direction amplification $L_{\text{core}}$.** The term $L_{\text{spur}}$ not only measures the magnitude of the spurious direction $\Delta_{\text{spur}}$ but also accounts for how much the classifier relies on this direction, ensuring robustness.
> Third, by incorporating classifier weights directly into $L_{\text{spur}}$, SCER ensures that the reduction of spurious signal is reflected in the decision boundary rather than only in the embedding geometry.
>
> In summary, Our method leverages the neural collapse framework and extends it with classifier-aware regularization to address spurious correlations, offering a principled approach to group fairness under severe data imbalance. We hope this clarifies the connection and distinctions. Thank you again for the thoughtful question.
>
> [7] Papyan, Vardan, X. Y. Han, and David L. Donoho. "Prevalence of neural collapse during the terminal phase of deep learning training." Proceedings of the National Academy of Sciences 117.40 (2020): 24652-24663.
>
> [8] Hong & Ling. Neural collapse for unconstrained feature model under cross-entropy loss with imbalanced data. arXiv:2309.09725, 2023.
>
> [9] Lu, Shenyu, Junyi Chai, and Xiaoqian Wang. "Neural collapse inspired debiased representation learning for min-max fairness." Proceedings of the 30th ACM SIGKDD Conference on Knowledge Discovery and Data Mining. 2024.
>
>
> > Q3. grammatical errors
>
> Thank you for taking the time to review our work and provide such constructive feedback. We addressed all the grammatical issues you identified. The corrections are marked in **blue throughout the revised paper**. The figures have also been updated, with corresponding changes made to the relevant text.
>
> Once again, we sincerely thank the reviewer for their time and thoughtful feedback. The comments have been extremely helpful in improving the clarity, logical coherence, and overall quality of our paper.

---

> > ### Comment · Reviewer_vc9T · 2025-11-26
> >
> > I appreciate the author's comprehensive responses. My questions (Q1) and (Q2) are satisfied; I recommend including the discussion (Q1) in the paper to contextualize the contributions to the understanding of embedding space dynamics under spurious correlation.
> >
> > I would like to respond to each of the weaknesses individually:
> >
> > (W1) I appreciate the modifications to the paper and I believe the comparisons are much improved. Please include GroupDRO-ES in each results table where GroupDRO is listed.
> >
> > (W2) I appreciate the GroupDRO-ES comparisons; the results are mostly as I expected in (W2) with the exception of CivilComments, upon which the dataset clarification is appreciated. Please clarify the 16-group vs 4-group versions of CivilComments in the paper in Appendix B.1.1 and distinguish which version was evaluated by each method in the tables. Overall, since SCER tunes one additional hyperparameter compared to GroupDRO-ES on validation set WGA, the 1-2% gain on Waterbirds, CelebA, and CivilComments is not very significant or surprising. However, the +4.4% improvement on MultiNLI is impressive, as to my knowledge most group robustness methods struggle to get past the low-70s WGA on this dataset. I will take this final point into consideration when making my final recommendation.
> >
> > (W3a) While the perturbative expansion is mathematically correct, it is specific to linear models, and therefore does not address my main concern that the shared feature covariance assumption does not hold in the neural networks which execute the SCER algorithm (as noted by previously mentioned empirical work). Unless the authors can show why the neural covariance structure behaves analogously to the linear model, I would prefer the paper to clearly present Theorem 1 as a motivation rather than a justification for SCER, and to refrain from implying direct applicability beyond the linear setting. In particular, I believe the statement that Theorem 1 provides "rigorous justification" or a "rigorous theoretical framework" for SCER is overclaimed.
> >
> > (W3b) The extreme correlation experiments are welcome. I would also appreciate a line or two of justification for the extreme correlation assumption in the theory (e.g., at the beginning of section 3.3).

---

> ### Author Response · Authors · 2025-12-01
>
> Thank you for your thoughtful feedback and for acknowledging the improvements we have made to the paper. We sincerely appreciate your constructive engagement throughout the review process.
>
> > **Q1. Performance comparison with GroupDRO**
>
> We have incorporated the discussion you suggested into the paper. **Figure 3** includes additional analysis, and the **Embedding Space Analysis section** provides a detailed description of the experimental results, **highlighted in red**. We are grateful for this valuable suggestion.
>
> > **W1. Baselines (Include GroupDRO-ES)**
>
> We have added comparisons with GroupDRO-ES to the paper. Following the original paper [1], we compared performance on selected datasets and used their reported results directly to ensure accuracy. **Since the experimental setup and dataset configuration differ from our other baselines**, we created a separate section and **present these results in Table 16**. We have provided detailed descriptions of the datasets and clarified the source of performance metrics to minimize potential confusion and clearly demonstrate our method's contributions. The **discussion from Q1 has been fully incorporated into this section (highlighted in red)**. Additionally, the **baseline section in the main text now references this supplementary experiment (highlighted in red)**. We plan to conduct additional experiments with aligned experimental settings and incorporate the results into the main text.
>
> [1] Izmailov, Pavel, et al. "On feature learning in the presence of spurious correlations." Advances in Neural Information Processing Systems 35 (2022): 38516-38532.
>
> > **W2. Comparison with GroupDRO**
>
> Our method demonstrates consistent performance improvements across diverse datasets, with particularly meaningful gains on the challenging MultiNLI benchmark, as shown in **Table 16 and highlighted in red**. We have **added clarification about the 4-group CivilComments setting in Table 8** and the accompanying text to reduce confusion and enable more rigorous comparison.
>
> Additionally, we note that the CivilComments (4 groups) performance improved from 82.5% in our initial response to **82.9%** after further hyperparameter tuning. While tuning remains incomplete, this suggests our method has even greater potential given sufficient time and resources. We deeply appreciate your suggestion for this rigorous comparison.
>
> > **W3.a. Assumption about shared covariance setting**
>
> We have revised the paper as you recommended. In the main text, **"theoretical justification" has been changed to "theoretical motivation" and is highlighted in red**. Additional revisions reflecting this framing have been made in the **main text and Appendix, also highlighted in red and green**.
>
> > **W3.b. Assumption about extreme correlation setting**
>
> Following your advice, we have added a remark in the Appendix to address this point within space constraints. We cite prior work to provide a brief justification for the extreme correlation assumption in the theoretical analysis and note that this is empirically validated, as shown on **Appendix page 4 and highlighted in red**.
>
>
> We are truly grateful for your careful, thorough review and the valuable suggestions you have provided throughout this process. Your insights have been instrumental in strengthening both the theoretical framing and empirical validation of our work. We especially appreciate your patience and willingness to engage deeply with our revisions, which has helped us present our contributions more clearly and rigorously. Thank you once again for your time, expertise, and thoughtful consideration of our paper.

---

> ### Author Response · Authors · 2025-12-01
>
> We sincerely thank the reviewer for their thorough and detailed review. The added comparison with GroupDRO, which validates the embedding regularization effects, and examination of the theoretical analysis assumptions directly address the important issues you raised. We are deeply grateful for your constructive feedback, which has significantly strengthened our work. We would also welcome any further discussion or questions about these revisions, should the review process allow.

---

### Official Review · Reviewer_4dbm · 2025-10-22

**Soundness:** 3
**Presentation:** 3
**Contribution:** 3
**Rating:** 8
**Confidence:** 4

**Summary:**

This paper proposes a novel method for mitigating spurious correlations by developing clear constraints on the embeddings learned from different subpopulation groups. The core feature is calculated as the average difference of embeddings from samples with the same domain but different labels, while the spurious feature considers the difference for samples with the same label but different domains. Next, the alignment between classifier weights and the directions of the normalized features is calculated and used as an additional loss (constraint) alongside the common worst-group classification loss. Experiments are conducted on popular datasets for studying spurious correlations, and the proposed method is compared with many baseline methods. The results demonstrate that the proposed method works as designed and is very promising.

**Strengths:**

(S1) The proposed constraints on the embeddings are explicit and look novel to me, especially the correlation terms defined in Eq. (1), which are well supported by Theorem 1. I believe the research directions enlightened by this paper are very promising. Besides, the Introduction and Related Works sections also provide a good discussion about this paper's contributions, which places it well among the literature on spurious correlations.

(S2) Comprehensive experimental settings provide strong evaluations of the performance of the proposed methods, although some improvements are marginal in Table 1. In general, I believe the evidence is enough to demonstrate that the proposed method is very promising. In particular, I really appreciate the results in Figure 3, which demonstrate that the proposed constraints on the correlation terms are working and work as designed.

(S3) In general, this paper is well written and easy to follow.

**Weaknesses:**

(W1) While a schematic diagram/representation of the embeddings learned from the proposed methods is presented in Figure 1, a comprehensive evaluation of the structure and clusters of the learned embeddings, in comparison to baseline methods, is not included in this paper, which weakens the evaluation of a key contribution of this paper: the explicit constraints on the embeddings. In particular, it would be very helpful for readers to examine the differences between the embeddings learned by the proposed methods and those learned by works focusing on learning invariant features.

**Questions:**

(Q1) The authors are encouraged to respond to (W1) I described above. Additional simple experimental results in the appendix would be much appreciated and could have a big impact on my final opinion, such as figures of t-SNE plots labeling different subpopulations, especially in comparison to baseline methods that also operate in the embedding space and focus on invariant features.

(Q2) (Minor) In Figure 1, instead of using mathematical notations for the proposed core/spurious features, plain language descriptions might improve readability. Personally, I had a difficult time understanding that part before reading the Methods section.

---

> ### Author Response · Authors · 2025-11-21
>
> Thank you for the valuable feedback. We conducted additional experiments and revised the paper to improve clarity. Below, we address each comment in detail.
>
> > W1, Q1. Embedding visualization comparison
>
> We have included **Figure 3 highlighted in blue in the revised paper** which presents t-SNE visualizations of embeddings learned by ERM, GroupDRO, and our proposed SCER method on the Waterbirds dataset. The t-SNE analysis reveals distinct characteristics of each method's learned representations.
>
> - **ERM** : The embeddings form clusters that are driven primarily by background rather than by label.
> Waterbird samples separate into two distant groups depending on background, and Landbird samples show the same pattern. This indicates that the model relies heavily on domain-specific cues instead of learning the underlying label structure.
> - **GroupDRO** : The model reduces but does not fully eliminate background-based separation. Samples with the same label still split into distinct clusters when their backgrounds differ, suggesting that GroupDRO only partially suppresses reliance on non-essential domain information despite its worst-group optimization.
> - **SCER** : The embeddings become clearly organized according to label, enabled by SCER’s explicit embedding regularization. Waterbird samples form a unified cluster regardless of background, and Landbird samples behave similarly. The two label clusters are well separated, providing qualitative evidence that SCER learns domain-invariant representations that emphasize core label-related features.
>
> We hope this visualization helps resolve your concern by illustrating that our explicit embedding constraints yield a clearer and more robust structure than approaches that rely on implicitly learning invariant features.
>
> > Q2. Readability of Figure 1
>
>
> Thank you for your valuable feedback. Following your suggestion, we have revised the Figure 1 caption to make the key components more intuitive. We've added plain language explanations for technical terms like Spurious Feature, Core Feature, Spurious Magnitude, Core Magnitude, Weight-Spurious Alignment, and Weight-Core Alignment. This should help readers understand the overall structure and concepts directly from the figure and caption, without needing to consult the Methods section. The revisions are **highlighted in blue in the Figure 1 caption** of the revised manuscript.
>
>
> Once again, we sincerely thank the reviewer for their time and thoughtful feedback. The comments have been extremely helpful in improving the clarity, logical coherence, and overall quality of our paper.

---

> > ### Author Response · Authors · 2025-12-01
> >
> > We sincerely thank the reviewer for their thorough and detailed review. The added visualizations and analysis of the embedding regularization effects directly address the important issues you raised. We are deeply grateful for your constructive feedback, which has significantly strengthened our work. We would also very much welcome any further discussion or questions about these revisions, should the review process allow.

---

### Official Review · Reviewer_NQJE · 2025-10-25

**Soundness:** 3
**Presentation:** 3
**Contribution:** 2
**Rating:** 4
**Confidence:** 4

**Summary:**

The paper proposes a new method to improve worst-group robustness by encouraging models to rely on core features while suppressing spurious correlations in the embedding space. It introduces a regularization term that rewards alignment with class-discriminative directions and penalizes reliance on non-causal relations. Experiments on vision and language benchmarks show consistent gains in worst-group accuracy over existing methods.

**Strengths:**

- The method is concisely proposed with theoretical analysis
- The results are clearly explained with interpretable evidence

**Weaknesses:**

- The core idea is to encourage models to rely less on spurious correlations and more on invariant features. This has been explored in a substantial body of prior work both empirically and theoretically. The contribution here, while sound, does not clearly break new conceptual ground, considering the improvements over baselines appear modest.
- I believe this work, as well as similar ones, should be applied to datasets with a larger number of groups. For example, where the number of groups is greater than 50, the spurious correlations are not very obvious to humans under the given group divisions.

**Questions:**

What do you think are the potential reasons for ElRep’s significant performance drop on MetaShift and ColorMNIST?

---

> ### Author Response · Authors · 2025-11-21
>
> Thank you for the valuable feedback. We conducted additional experiments and revised the paper to improve clarity. Below, we address each comment in detail.
>
> > W1. Novelty of SCER
>
> Thank you very much for your thoughtful feedback and for raising this important point regarding the existing body of work on spurious correlations. We greatly appreciate your engagement with our manuscript and the opportunity to clarify the distinctive contributions of our work.
>
> We fully acknowledge that research on spurious correlations is well-established and widely recognized. However, we argue that SCER offers qualitatively distinct contributions through its novel theoretical foundation and direct embedding space manipulation, which clearly differentiate it from existing approaches.
>
> First and foremost, SCER does not simply iterate upon existing empirical strategies that aim to "reduce spurious signals and strengthen invariant features." Rather, it provides a rigorous mathematical decomposition of worst-group error in terms of structural factors within the embedding space. Building on this theoretical framework, we introduce a novel objective function that explicitly separates and controls two components: the 'spurious component' and the 'core component.' In other words, while existing methods have pursued directions that consequentially reduce spurious correlations, SCER fundamentally defines the root cause of worst-group error from an embedding geometry perspective and directly controls it. This constitutes the core theoretical novelty of our work.
>
> Second, the practical strength of SCER lies in its ability to maintain stable performance even under extreme subpopulation shift scenarios. To demonstrate this more clearly in the revised manuscript, we have added new experiments **highlighted in blue in Table 2** with ColorMNIST at $\rho=99\\%$. This represents an extreme condition where the spurious correlation between domain and label is nearly perfect. Under these challenging circumstances, most existing methods exhibit severe worst-group performance collapse, whereas SCER maintains remarkably robust worst-group accuracy.
>
> In summary, we believe the key contributions of SCER are not merely about performance improvements, but rather about establishing a qualitatively different direction of novelty through three distinct pillars:
>
> 1. **Theoretical foundation**: We provide the first rigorous theoretical connection between worst-group error and embedding geometry.
> 2. **Methodological innovation**: We propose an explicit decomposition and control mechanism for spurious and core components in the embedding space.
> 3. **Empirical robustness**: We demonstrate stability even under extreme spurious correlation conditions, which represents a significant advancement over existing methods.

---

> ### Author Response · Authors · 2025-11-21
>
> > W2. Scalability of groups
>
> Thank you for suggesting validation on datasets with more groups. We agree this is important for assessing practical applicability, so we conducted additional experiments on the CelebA dataset.
>
> Beyond the Male attribute commonly used in existing studies, we selected four additional attributes that appear to have lower direct correlation with the Blond Hair class. This was designed to construct a more complex and subtle group structure that is difficult for humans to intuitively identify. Through the combination of five attributes, a total of $2^5$ attributes are generated, and when the class label is included, this results in more than 50 subgroups.
>
> ### [Table 1] Performance of SCER on CelebA under a large number of groups
>
> | Model | Avg ACC | Worst ACC |
> |-------|---------|-----------|
> | ERM   | 92.4 ±0.3    | 0.0 ±0.0      |
> | DRO   | 70.6 ±19.0 | 27.8 ±12.0 |
> | SCER  | 89.4 ±0.4 | 50.0 ±7.9 |
>
> In this challenging setting, some groups contain extremely sparse samples—certain groups have only 1 or 3 instances. As shown in Table 1, ERM completely fails on the worst group despite high average performance, while DRO improves worst-group accuracy to 27.8% but severely degrades average accuracy. In contrast, SCER achieves strong average accuracy while nearly doubling DRO's worst-group performance, demonstrating robust performance even under extreme group fragmentation and sparsity.
>
> We believe these additional results strongly support the robustness and scalability of SCER in real-world scenarios.
>
>
> > Q1. ElRep’s significant performance drop on MetaShift and ColorMNIST
>
> Thank you for raising this important point regarding the comparison with ElRep. We would like to respectfully clarify the differences in experimental settings and their potential impact on performance.
>
> For fair and consistent comparison, we have carefully followed the experimental protocol established in the prior benchmark study [1], maintaining the 5,000 training step configuration for ColorMNIST and MetaShift.
>
> ElRep [2] employs a dual-regularization structure that simultaneously applies nuclear norm and Frobenius norm to the representation matrix. According to prior studies, such composite low-rank regularization creates a more complex optimization landscape compared to simple Frobenius norm-based regularization, thereby slowing convergence [3,4]. In particular, Peng et al. [3] demonstrated that when nuclear norm and Frobenius norm are combined, the shape of the objective function varies depending on dictionary capacity, which increases optimization difficulty.
>
> Therefore, given the limited 5,000step training in ColorMNIST and MetaShift, it is possible that ElRep's dual-regularization structure prevented full convergence, which could partly explain the performance differences observed in our study.
>
> We hope this fully addresses your concern.
>
>
> [1] Yang, Yuzhe, et al. "Change is hard: a closer look at subpopulation shift." Proceedings of the 40th International Conference on Machine Learning. 2023.
>
> [2] Wen, Tao, et al. "Elastic Representation: Mitigating Spurious Correlations for Group Robustness." The 28th International Conference on Artificial Intelligence and Statistics.
>
> [3] Peng, Xi, et al. "Connections between nuclear-norm and frobenius-norm-based representations." IEEE transactions on neural networks and learning systems 29.1 (2016): 218-224.
>
> [4] Guo, Huiyuan, et al. "LOW RANK MATRIX MINIMIZATION WITH A TRUNCATED DIFFERENCE OF NUCLEAR NORM AND FROBENIUS NORM REGULARIZATION." Journal of Industrial & Management Optimization 19.4 (2023).
>
>
> Once again, we sincerely thank the reviewer for their time and thoughtful feedback. The comments have been extremely helpful in improving the clarity, logical coherence, and overall quality of our paper.

---

> > ### Comment · Reviewer_NQJE · 2025-11-27
> >
> > Thank you to the authors for the explanation of ElRep.
> >
> > I have some doubts about the performance of GroupDRO. From my past experience and according to past research, GroupDRO usually does not perform this poorly on CelebA and data with a large number of groups. For example, in the WILDS benchmark, even if GroupDRO may sometimes underperform ERM, having many groups typically does not cause such a dramatic drop in performance.
> >
> > What is causing GroupDRO to degrade so severely in your setting when many manually constructed groups are added? What does the distribution of group sizes look like in your setup? Is it possible that the presence of very small groups is introducing bias into the algorithm?

---

> > > ### Author Response · Authors · 2025-12-01
> > >
> > > Thank you for your thoughtful feedback and for acknowledging the improvements we have made to the paper. We sincerely appreciate your constructive engagement throughout the review process.
> > >
> > > > Additional Q1. What is causing GroupDRO to degrade so severely in large group setting ?
> > >
> > > The presence of very small groups is indeed the primary cause of GroupDRO's degradation in our setting. To ensure a fair comparison, we conducted all experiments following the standardized protocol described in our main text, which is consistent with the evaluation framework proposed in [1]. All methods were evaluated under identical conditions.
> > >
> > > As shown in **Table 1 in the comments** (Performance of SCER on CelebA under a large number of groups), our experimental setting differs from standard WILDS-style benchmarks in how groups are constructed. In our experiments, groups are defined by intersecting multiple manually specified attributes, which leads to a much more fine-grained partition of the data. This results in a large number of groups with highly imbalanced sizes, creating a long tail of very small groups.
> > >
> > > As demonstrated in **Additional Table 1 below**, our CelebA setting uses 32 attributes instead of 2. While the original CelebA has a minimum group size of 1,387 samples, ours includes groups with as few as 7 samples.
> > > **Additional Tables 2 and 3** show the top-5 largest and smallest groups. The largest group contains 22,058 training samples while the smallest has only 7—over three orders of magnitude difference. **Additional Table 4 provides the complete distribution**, further demonstrating this extreme long-tailed imbalance.
> > >
> > > ### [Additional Table 1] Comparison with original CelebA
> > >
> > > | Dataset        | Data type | # Attr. | # Classes | # Train | # Val. | # Test | Max group | Min group |
> > > |----------------|-----------|---------|-----------|---------|---------|---------|-----------|-----------|
> > > | CelebA         | Image     | 2       | 2         | 162,770 | 19,867  | 19,962  | 71,629    | 1,387     |
> > > | **CelebA (large number of groups)** | Image     | 32    | 2         | 162,770 | 19,867 | 19,962 | 22,058  | 7      |
> > >
> > > ### [Additional Table 2] Top-5 largest distribution of (y, a) Groups Across Train, Validation, and Test Splits on CelebA under a large number of groups
> > >
> > > | rank | y | a  | train | valid | test | total |
> > > |------|---|----|-------|-------|------|-------|
> > > | 1    | 0 | 30 | 22,058 | 2,442  | 3,132 | 27,632 |
> > > | 2    | 0 | 4  | 20,929 | 2,514  | 2,452 | 25,895 |
> > > | 3    | 0 | 5  | 19,652 | 2,454  | 2,172 | 24,278 |
> > > | 4    | 0 | 1  | 9,815  | 1,198  | 1,200 | 12,213 |
> > > | 5    | 0 | 31 | 8,268  | 975   | 948  | 10,191 |
> > >
> > > ### [Additional Table 3] Top-5 smallest distribution of (y, a) Groups Across Train, Validation, and Test Splits on CelebA under a large number of groups
> > >
> > > | rank | y | a  | train | valid | test | total |
> > > |------|---|----|-------|-------|------|-------|
> > > | 1    | 1 | 25 | 7     | 3     | 2    | 12    |
> > > | 2    | 1 | 13 | 15    | 1     | 1    | 17    |
> > > | 3    | 1 | 9  | 13    | 2     | 3    | 18    |
> > > | 4    | 1 | 29 | 14    | 1     | 3    | 18    |
> > > | 5    | 1 | 3  | 30    | 0     | 6    | 36    |
> > >
> > > While GroupDRO has been influential in robust optimization [2], it faces several practical limitations that become critical when very small groups are present:
> > >
> > > 1. **Extreme minority group instability**: When minority groups contain only a handful of samples (e.g., 7 training samples, 2 test samples), GroupDRO's reweighting scheme becomes highly unstable. The group-specific losses on these tiny groups have extremely high variance, causing the reweighting mechanism to behave erratically.
> > >
> > > 2. **Near-unseen group behavior**: Groups with extremely limited training data behave nearly **as unseen subpopulations**. GroupDRO's uncertainty set is defined as the convex hull of the group distributions $P_g$ observed in the training data, meaning it considers worst-case scenarios only within the range of mixing proportions among **adequately represented** groups. When a group has insufficient samples, it effectively falls outside this framework.
> > >
> > > 3. **Regularization-capacity trade-off**: To achieve stability with very small groups, GroupDRO would require extremely strong regularization, which significantly reduces model capacity and hurts overall performance.
> > >
> > > **In our dataset setting, many groups have extremely limited training data—some with as few as 7 training samples (e.g., (y,a)=(1,25)) and only 2 test samples. As you correctly suspected, these very small groups introduce significant instability into GroupDRO's reweighting mechanism, causing the severe performance degradation we observed.**
> > >
> > > This is distinct from typical WILDS benchmarks where, even with many groups, the smallest groups still contain sufficient samples for stable optimization. Our setting represents an extreme case of group imbalance that exposes fundamental limitations of reweighting-based approaches when groups are severely underrepresented.

---

> > > > ### Author Response · Authors · 2025-12-01
> > > >
> > > > This evidence supports that while GroupDRO can be effective in moderate bias scenarios, embedding-level interventions become necessary under extreme spurious correlations where reweighting alone cannot compensate for severely corrupted representation spaces that fail to encode sufficient core feature information.
> > > >
> > > > ### [Additional Table 4] Full distribution of (y, a) Groups Across Train, Validation, and Test Splits on CelebA under a large number of groups
> > > >
> > > > | y | a | train | valid | test |
> > > > |---|---|-------|-------|------|
> > > > |0|0|1,220|234|219|
> > > > |0|1|9,815|1,198|1,200|
> > > > |0|2|114|19|25|
> > > > |0|3|1,315|137|149|
> > > > |0|4|20,929|2,514|2,452|
> > > > |0|5|19,652|2,454|2,172|
> > > > |0|6|1,326|154|175|
> > > > |0|7|2,032|250|220|
> > > > |0|8|493|86|102|
> > > > |0|9|1,045|139|146|
> > > > |0|10|1,006|150|195|
> > > > |0|11|1,746|225|163|
> > > > |0|12|3,665|357|529|
> > > > |0|13|1,295|108|120|
> > > > |0|14|6,428|712|852|
> > > > |0|15|1,751|200|201|
> > > > |0|16|537|95|94|
> > > > |0|17|2,591|301|294|
> > > > |0|18|312|67|43|
> > > > |0|19|2,198|321|258|
> > > > |0|20|6,236|729|787|
> > > > |0|21|4,637|551|473|
> > > > |0|22|2,288|240|293|
> > > > |0|23|4,129|479|470|
> > > > |0|24|211|28|38|
> > > > |0|25|476|73|71|
> > > > |0|26|3,700|594|681|
> > > > |0|27|5,166|776|589|
> > > > |0|28|1,106|114|150|
> > > > |0|29|758|89|61|
> > > > |0|30|22,058|2,442|3,132|
> > > > |0|31|8,268|975|948|
> > > > |1|0|519|84|81|
> > > > |1|1|176|22|22|
> > > > |1|2|43|7|10|
> > > > |1|3|30|0|6|
> > > > |1|4|5,147|641|501|
> > > > |1|5|350|50|28|
> > > > |1|6|358|49|34|
> > > > |1|7|44|6|9|
> > > > |1|8|242|23|44|
> > > > |1|9|13|2|3|
> > > > |1|10|450|77|58|
> > > > |1|11|32|3|1|
> > > > |1|12|1,006|115|109|
> > > > |1|13|15|1|1|
> > > > |1|14|1,992|273|250|
> > > > |1|15|35|4|2|
> > > > |1|16|208|23|22|
> > > > |1|17|45|8|9|
> > > > |1|18|123|13|23|
> > > > |1|19|50|9|3|
> > > > |1|20|1,832|171|175|
> > > > |1|21|113|13|16|
> > > > |1|22|644|88|55|
> > > > |1|23|112|14|26|
> > > > |1|24|69|4|11|
> > > > |1|25|7|3|2|
> > > > |1|26|1,920|291|270|
> > > > |1|27|110|23|16|
> > > > |1|28|268|34|40|
> > > > |1|29|14|1|3|
> > > > |1|30|8,059|981|797|
> > > > |1|31|241|23|33|
> > > >
> > > >
> > > > [1] Yang, Yuzhe, et al. "Change is hard: a closer look at subpopulation shift." *Proceedings of the 40th International Conference on Machine Learning*. 2023.
> > > >
> > > > [2] Sagawa, Shiori, et al. "Distributionally Robust Neural Networks." *International Conference on Learning Representations*.
> > > >
> > > >
> > > > We sincerely appreciate your thorough review and valuable feedback, which have been instrumental in strengthening our work. Thank you for your time and expertise.

---

> ### Author Response · Authors · 2025-12-01
>
> We sincerely thank the reviewer for their thorough and detailed review. The additional experiments on larger groups, the discussion with other models, and the emphasis on the novelty of our approach directly address the important issues you raised. We are deeply grateful for your constructive feedback, which has significantly strengthened our work. We would also very much welcome any further discussion or questions about these revisions, should the review process allow.

---

### Official Review · Reviewer_mZvF · 2025-10-31

**Soundness:** 3
**Presentation:** 3
**Contribution:** 3
**Rating:** 6
**Confidence:** 3

**Summary:**

This paper proposes Spurious Correlation-Aware Embedding Regularization (SCER) as a method to improve on worst-group robustness by proposing embedding-level regularization. The authors provide theoretical justification for decomposing worst-group error into spurious and core components. The method identifies spurious directions as differences in embeddings across domains within the same class, while core directions are differences across classes within the same domain and combines them into the SCER objective that penalize classifier alignment with the spurious direction while encouraging alignment with the core direction. Experiments on vision and language datasets show improvements over existing methods, particularly in extreme settings where a subpopulation is completely missing during training.

**Strengths:**

The paper precisely defines core and spurious mean-embedding differences, aggregates them into global directions, and integrates their alignment with classifier weights into a composite loss term that is transparent and straightforward to implement.

The authors conduct comprehensive experiments 6 datasets across vision and language domains, comparing against 19 baseline methods. The settings also cover different spurious correlation strengths and extreme scenarios.

Notably, the method substantially outperforms baselines when one subpopulation is entirely absent during training on ColorMNIST.
The authors conduct sensitivity and component analyses to support specific design choices and validate their theoretical decomposition.

**Weaknesses:**

When formulating the theorem for error decomposition, the paper is restricted to binary labels and two domains and assumes constant core and spurious differences between domains and classes. The paper does not discuss how violations of these assumptions affect the method's validity, particularly for multi-class problems, which appear in experiments like MultiNLI with 3 classes.

It can be argued that correctly specified group labels are required to operationalize SCER in terms of its component losses. While this is briefly explored with EIIL in a two-environment setup, it would be important to understand how SCER behaves when group labels are noisy. For example, how does robustness vary when EIIL’s inferred groups are partially misaligned to 70-80% agreement instead of 95%?

The paper computes group-wise mean embeddings but doesn't discuss key considerations, such as how many samples are needed for stable estimates, what happens when group sizes are highly imbalanced, and what happens if spurious and core features are not well-separated in embedding space.

Furthermore, aggregating across all classes or groups into a single spurious and single core direction may lead to underfitting in settings with multiple heterogeneous spurious factors. A discussion and validation of how this is handled is missing.

**Questions:**

Would it be possible to discuss how extending the error decomposition theorem beyond the two-class/groups assumption can be justified for the sake of generality?

How does SCER behave in terms of robustness and performance when group labels are noisy, for instance, when inferred groups are partially misaligned to 70-80% agreement instead of 95%?

How would you handle when multiple spurious or core directions exist? Can you provide evidence that consolidating into a single global direction is optimal?

How many training steps does SCER typically require compared to baselines? The per iteration time is provided, but not the full training cost

---

> ### Author Response · Authors · 2025-11-21
>
> Thank you for the valuable feedback. We conducted additional experiments and revised the paper to improve clarity. Below, we address each comment in detail.
>
> > W1, Q1.  how our binary/two-domain error decomposition extends to multi-class or multi-group settings like MultiNLI.
>
>
> Thank you for this important question regarding generalizability. While our theoretical framework is initially presented in a 2-class, 2-domain setting, we address this **generalization concern in Appendix A.2.2 in the revised paper highlighted in blue** by extending the analysis to multi-class and multi-domain scenarios.
>
> In this extension, we construct conditional Gaussian models for each class pair and leverage the fact that the decision boundaries of a multiclass softmax classifier can be represented as a collection of pairwise linear boundaries $\beta_i - \beta_j$. We demonstrate that the fundamental structure derived from the binary setting is preserved for each class pair in the multiclass case. Specifically, core direction alignment increases classification margin while spurious direction alignment decreases it. Furthermore, we theoretically establish that the worst-group error in multiclass settings can be controlled by the upper and lower bounds of pairwise margins across all class pairs, thereby justifying that our binary analysis remains structurally valid in multiclass scenarios.
>
> These theoretical results explain why SCER demonstrates robust performance in practical experiments involving multiclass problems such as MultiNLI and CivilComments.
>
> > W2, Q2. SCER’s robustness when group labels are noisy or imperfectly inferred.
>
>
> To address your question, we conducted additional analysis in a two-environment setting using group labels inferred by EIIL. Rather than relying on the near-perfect alignment (≈95%) provided by EIIL, we artificially reduced the group alignment accuracy to 70–80%, as you suggested, to better reflect realistic noisy scenarios. All other environmental configurations and experimental settings remain identical to those in Table 4 of our paper.
>
> Our experimental results demonstrate that even when group label accuracy drops to approximately 70%, SCER maintains significantly higher performance compared to IRM, EIIL, and EIIL+DRO. The **detailed results are presented in Table 14 of the revised paper and highlighted in blue**. Specifically, EIIL + SCER achieves 65.1  ± 2.8% average accuracy, substantially outperforming EIIL + DRO at 54.6 ± 5.9%, EIIL with IRM at 53.7 ± 2.1%, and baseline IRM at 54.8 ± 8.3%. These findings indicate that SCER's robustness does not depend on perfect group labels, and the method can stably capture useful invariant structures even in the presence of substantial noise.
>
> In summary, these results confirm that SCER is robust to realistic levels of group label noise and can achieve substantial performance gains even with partially aligned group information.

---

> ### Author Response · Authors · 2025-11-21
>
> > W3. the reliability of group-wise mean embeddings, especially under small, imbalanced, or poorly separated groups.
>
>
> Thank you for raising this important concern about SCER's robustness under limited sample conditions. We address this through both theoretical analysis and additional experiments.
>
> Under the Gaussian mixture model assumption, the group-wise mean embedding $\hat{\mu_{y,d}}$ is an unbiased estimator of the true mean $\\mu^{(y,d)}$, with estimation error proportional to $\\Sigma / n_{y,d}$ The variability of the mean embedding decreases at the typical rate of $O(1/\\sqrt{n_{y,d}})$ as the sample size increases. Since SCER's core and spurious directions are defined as differences between two group means, their estimation error follows the form $\frac{1}{n_a} + \frac{1}{n_b}$, where $n_a$ and $n_b$ are the sample sizes of the two groups (e.g., different labels or domains) defining the direction. This matches the classical standard error formula for the difference of two sample means, resulting in estimation noise that decreases as $O(1/\sqrt{n_{\mathrm{eff}}})$, where $n_{\mathrm{eff}}^{-1} = n_a^{-1} + n_b^{-1}$. Consequently, SCER does not require an absolute threshold like "each group must have at least N samples." Instead, it only requires that the estimation error not be so large as to obscure the meaningful differences that core or spurious directions actually possess. This perspective aligns with existing results [1] showing that mean embedding estimators converge at $O(1/\sqrt{n})$ under mild conditions. However, it is important to note that if either group has very few samples, the variance of the core/spurious direction estimates increases sharply, and the smaller group dominates the overall uncertainty.
>
> To address this concern more thoroughly, we conducted additional experiments on ColorMNIST with $\rho = 99\\%$, which represents an almost complete spurious correlation under extremely challenging conditions. These results are reflected in **Table 2 of the revised manuscript (highlighted in blue)**. Even in this highly demanding setting, while most existing methods experience severe degradation in worst-group performance, SCER maintains consistent worst-group accuracy, demonstrating exceptional practical robustness even under extreme subpopulation shift scenarios. We observed that as the spurious correlation becomes stronger, the variance also tends to increase. However, compared to other baselines, SCER maintains robust worst-group accuracy and keeps the variance at a reasonable level.
>
>
> Furthermore, to evaluate SCER under even more extreme conditions where minority groups have very few samples and numerous attributes make individual groups difficult to distinguish, we conducted additional experiments based on the CelebA dataset. Beyond the Male attribute commonly used in existing studies, we selected four additional attributes that appear to have lower direct correlation with the Blond Hair class. Through the combination of five attributes, a total of $2^5$ attribute combinations are generated, which, when combined with the class label, results in more than 50 subpopulations.
>
> **[Table 1] Performance of SCER on CelebA under a large number of groups**
>
> | Model | Avg ACC | Worst ACC |
> |-------|---------|-----------|
> | ERM   | 92.4 ±0.3 | 0.0 ±0.0 |
> | DRO   | 70.6 ±19.0 | 27.8 ±12.0 |
> | SCER  | 89.4 ±0.4 | **50.0 ±7.9** |
>
> Remarkably, even when some minority groups contain only 1-3 samples, SCER significantly outperforms other baselines in this extreme setting, maintaining not only superior worst-group accuracy but also notably lower variance.
>
> SCER leverages the statistical stability of direction estimation based on group mean differences, with a convergence rate of $O(1/\sqrt{n_{y,d}})$, ensuring robustness under extreme conditions. Experiments confirmed its stability even with complex group structures and severe imbalances.
>
> [1] Lerasle, Matthieu, et al. "Monk outlier-robust mean embedding estimation by median-of-means." International conference on machine learning. PMLR, 2019.

---

> ### Author Response · Authors · 2025-11-21
>
> > W4, Q3.Concern that collapsing multiple spurious factors into a single global direction may be unjustified
>
> Thank you for raising this important concern regarding the consolidation of multiple spurious factors into a single global direction.
>
> As shown in our methodology, SCER computes spurious and core directions through expectations across all class-domain combinations. Specifically:
>
> $$
> \Delta _ {\text{spur}} = \mathbb{E} _ {y \in \mathcal{Y}} \left[ \Delta _ {\text{spur}}^{(y, \, d _ {i,j})} \right], \quad
> \Delta _ {\text{core}} = \mathbb{E} _ {d \in \mathcal{D}} \left[ \Delta _ {\text{core}}^{(y _ {i,j}, \, d)} \right]
> $$
>
> As can be seen from these definitions, we compute the differences for all domain pairs within each class ($\Delta_{\text{spur}}^{(y,d_{i,j})}$) and average them across all classes. Similarly, the core direction is obtained by averaging differences across all class pairs within each domain, aggregated over all domains.
>
> The key insight is that this averaging process does not simply ignore or arbitrarily select one spurious factor among many. Rather, it comprehensively captures all spurious variability through group-wise mean embeddings. Our mean embedding-based approach preserves the representative characteristics of each subpopulation while enabling global utilization of the structural relationships among them.
>
> For empirical evidence, we provide Grad-CAM visualization results in **Figure 5 and 6 of the revised paper, which is highlighted in blue**. These results demonstrate that while ERM focuses primarily on background elements, and DRO captures core features only partially, **SCER comprehensively and globally captures elements that can be considered core features, such as the overall shape of the bird, its beak and other structural characteristics.** This shows that our mean embedding-based penalty approach effectively preserves diverse core features while successfully suppressing spurious factors.
>
> These findings suggest that consolidating into a single global direction does not result in information loss. Rather, it captures the common structure of multiple spurious patterns, thereby enabling robust generalization.
>
>
> > Q4. About training cost
>
> Thank you for your question regarding the total training cost of SCER compared to baselines.
>
> In our CMNIST experiments, we allowed a maximum of 5,000 training steps for all methods. For seed 0 under the "$\rho$ = 80" experimental setting reported in the paper, the actual number of steps required for convergence by each method was as follows:
>
> - ERM: 4500–5000 steps
> - DRO: 1250, 1750, 1000 steps
> - SCER: 750, 1500, 1000 steps
>
> These results show that while SCER has a slightly higher per-iteration cost, it converges much faster than both DRO and ERM. Similar trends appear across other datasets, where SCER converges at a comparable rate to DRO and rarely needs the maximum training steps that ERM requires. This highlights SCER's overall training efficiency.
>
> We hope this information adequately addresses your question.
>
>
> Once again, we sincerely thank the reviewer for their time and thoughtful feedback. The comments have been extremely helpful in improving the clarity, logical coherence, and overall quality of our paper.

---

> > ### Author Response · Authors · 2025-12-01
> >
> > We sincerely thank the reviewer for their thorough and detailed review. The added theoretical analysis, broader experiments on robustness, and the expanded discussion of the SCER directly address the important issues you raised. We are deeply grateful for your constructive feedback, which has significantly strengthened our work. We would also very much welcome any further discussion or questions about these revisions, should the review process allow.

---

### Author Response · Authors · 2025-11-21
**Global response**

We sincerely thank all reviewers (mZvF, NQJE, 4dbm, vc9T) for their thorough and valuable feedback, which has significantly improved the quality of our paper. This global response summarizes the key strengths highlighted across the reviews and the revisions made to address the concerns raised. Individual reviewer comments are addressed separately below.

### 1. Strengths of Our Work, as Highlighted by the Reviewers

1. [mZvF, NQJE, 4dbm, vc9T] **Comprehensive Experimental Validation** – Thorough comparisons across multiple domains and baselines with clearly explained and interpretable evidence.
2. [mZvF, 4dbm] **Transparent Integration of Core/Spurious Directions** – A precise, aggregated, and straightforward composite loss to address spurious correlations.
3. [NQJE] **Theoretical Soundness** – Well-motivated decomposition of worst-group error.
4. [vc9T] **Addresses a Critical Research Area** – Explores under-examined paths to group robustness.
5. [4dbm] **Writing Quality** – Clear and well-structured writing style.

### 2. List of Revisions

1. **Extended theoretical analysis** for multiclass and multidomain settings, including discussion of underlying conditions. (Response to mZvF)
2. **Enhanced qualitative analysis**, including t-SNE embeddings and Grad-CAM analyses for improved interpretability, confirming that SCER achieves domain-invariant clustering and concentrated activation on core features. (Response to mZvF, 4dbm)
3. **Demonstrated SCER's practical extensibility and robustness**, showing compatibility with other methods and maintaining strong performance under extreme bias, large group conditions, and noisy-label conditions. (Response to mZvF, NQJE, vc9T)
4. **Compared computational efficiency** between SCER and baseline methods. (Response to mZvF)
5. **Validated the importance of embedding regularization** through experimental comparisons with GroupDRO. (Response to vc9T)
6. **Justified theoretical assumptions**, providing clarification of our modeling conditions and discussing their connections to prior research. (Response to vc9T)
7. **Corrected figures and writing** for improved completeness and clarity. (Response to 4dbm, vc9T)

---

### Author Response · Authors · 2025-12-01
**[To New AC] Final response after discussion**

### **Dear New Area Chair,**

We understand that due to the recent incident, you have been newly assigned to our submission. We are deeply grateful for your time and effort in taking on this responsibility. We write to respectfully provide context for our paper and to highlight the substantial progress made during the discussion period.

### Paper Summary

Our research introduces Spurious Correlation Aware Embedding Regularization for Worst Group Robustness (SCER), a method that establishes a theoretical connection between worst-group error and embedding space structure under subpopulation shifts. SCER proposes a novel objective function that decomposes worst-group error into spurious and core components, then directly regularizes the embedding space to suppress spurious correlations while enhancing core features. **Reviewers recognized several key strengths of our approach, including comprehensive experimental validation, theoretical soundness, a new methodological perspective, and clear presentation.**

### Key Revisions & Discussion Outcome

**We thoroughly addressed all reviewer concerns with comprehensive revisions during the discussion period. After resolving initial concerns raised by reviewers, we provided additional experiments and clarifications to address their follow-up questions.**

- **For Reviewer mZvF**, we provided additional theoretical analysis and robustness experiments under noisy conditions.

- **For Reviewer NQJE**, we conducted extensive experiments across many groups and strengthened theoretical comparisons, **which resolved their initial concerns** and prompted follow-up questions.

- **For Reviewer 4dbm**, we added visualization results clarifying the method's behavior.

- **For Reviewer vc9T**, we added comparisons with the state-of-the-art baseline, GroupDRO-ES, and substantially improved the manuscript clarity. We also conducted the requested extreme correlation experiments and added theoretical justification, effectively addressing their concerns. **The reviewer acknowledged the strength of our MultiNLI results and indicated they would consider our responses in their final recommendation.**

### Specific Actions Taken

1. **Extended theoretical analysis** for multiclass and multidomain settings, demonstrating that our generalization guarantees hold even when certain assumptions are relaxed (Response to mZvF, vc9T).

2. **Enhanced qualitative analysis** with t-SNE embeddings and Grad-CAM visualizations, confirming that SCER achieves domain-invariant clustering and concentrated activation on core features (Response to mZvF, 4dbm).

3. **Demonstrated SCER's practical robustness** across extreme bias scenarios, large group conditions, and noisy-label settings, while showing compatibility with other methods (Response to mZvF, NQJE, vc9T).

4. **Validated the necessity of embedding regularization** through extensive comparisons with GroupDRO. During the discussion period, we conducted additional hyperparameter tuning and rigorous comparisons with the most competitive GroupDRO-ES baseline, further confirming our method's effectiveness. **The reviewer acknowledged the strength of our results, particularly on challenging benchmarks, and indicated they would consider this in their final recommendation** (Response to vc9T).

5. **Justified theoretical assumptions** by clarifying modeling conditions and connecting them to prior research. We also conducted additional experiments designed explicitly for extreme spurious correlation scenarios and provided theoretical justification, **which the reviewer welcomed and acknowledged positively** (Response to vc9T).

6. **Compared computational efficiency** between SCER and baseline methods (Response to mZvF).

7. **Improved presentation quality** by conducting additional experiments in response to reviewer questions, comprehensively revising the manuscript throughout, and clearly tracking **all changes with blue highlights for first-round comments** and **red and green highlights for second-round comments**, **demonstrating our active engagement and thorough commitment to addressing all reviewer feedback** (Response to mZvF, NQJE, 4dbm, vc9T).

### Conclusion

We believe SCER makes meaningful contributions to mitigating spurious correlations through its embedding regularization approach. Through active engagement during the discussion period, we have comprehensively addressed all reviewer concerns with additional theoretical analysis, extensive experiments, and substantial manuscript improvements. We respectfully ask for your consideration of our work's contributions to the field and the extensive revisions we have made in response to reviewer feedback.

Thank you for your time and consideration.

### Sincerely,

### The Authors

---

### Meta-Review · Area_Chair_1Pzr · 2026-01-07

**Summary:**

The paper proposes an extension of the Group DRO algorithm called SCER to improve robustness to spurious correlations. The method is based on feature regularization suppressing spurious cues. The paper provides both theoretical motivation and quite strong empirical results.

The reviewers recognize the novelty of the proposed algorithm, its performance and theoretical motivation. The major concerns raised included comparison to stronger baselines (in particular, Group DRO with early stopping) and robustness in the settings with multiple groups in the dataset as well as when group labels are not provided and have to be approximated, which were adequately addressed during the rebuttal.

**Reviewer Concerns:**

Concerns mostly addressed by the rebuttal:
- Result on datasets with more than two domains
- Robustness in inferred group setup (w/o true train group labels)

Outstanding concerns:
- The extent of technical novelty given prior literature in spurious correlations including feature based and regularization based approaches
- Gains on top of GroupDRO + ES

**Reviewer Scores:**

Reviewer mZvF raised concerns about the performance on datasets with multiple groups, robustness to group label approximation noise and low data regime. I believe these concerns were addressed in the rebuttal and the reviewer would retain score 6.

Reviewer NQJE requested experiments on datasets with multiple groups and baseline GroupDRO performance. I believe these questions were partially addressed during the rebuttal and the reviewer would either retain the score 4 or increase to 6 (less likely).

Reviewer NQJE requested additional analysis of the feature embeddings and their comparison to baseline methods but overall appreciated the work's novelty and empirical results so they would likely retain score 8.

Reviewer vc9T raised several concerns: 1) comparison to GroupDRO performance with early stopping (I believe this one is a major concern), 2) theoretical assumptions and their connection to the method, 3) clarity. The reviewer indicated that the rebuttal resolved most concerns, in particular the major concern regarding GroupDRO-ES and I believe this reviewer would be likely to increase the score to 4 or 6.

Given generally positive feedback from the reviewers I recommend “conditional acceptance” of the paper given:
1) The authors incorporate GroupDRO-ES results from the rebuttal into the final version
2) Additionally, the authors demonstrate that GroupDRO + ES + SCER has improvements on top of GroupDRO + ES (currently the comparison is between GroupDRO + ES vs GroupDRO + SCER).

---

### Decision · Program_Chairs · 2026-01-26

Accept (Poster)